# Pangolin hunting in southeast Nigeria is motivated more by local meat consumption than international demand for scales

Charles A. Emogor [1,2,3] ✉, Samuel K. Wasser [4], Lauren Coad[5,6], Ben Balmford[7], Daniel J. Ingram [8], Amayaa Wijesinghe [9], Benedict A. Atsu [3], Frederick Bassey[3], Dominic S. Ogu[3], Ngozi Okafor [3] & Andrew Balmford [1]

Thousands of species are threatened by overexploitation, often driven by a complex interplay of local and global demand for various products—a dynamic frequently overlooked in wildlife trade policies. African pangolins, regarded as the world's most trafficked wild mammals, are a heavily exploited group for different reasons across geographic scales. However, it remains unclear how far the burgeoning trafficking of their scales to Asia for medicine drives their exploitation compared with local meat demand. Here, using data collected from questionnaires distributed to 809 hunters and meat vendors in Nigeria, the world's biggest hub for pangolin trafficking, we show that targeted pangolin hunts are uncommon in the country's largest pangolin stronghold. Instead, 97% of pangolins are captured opportunistically or during general hunting, with 98% of these caught for meat and mostly either eaten by hunters (71%) or traded locally (27%), potentially due to the meat's exceptionally high palatability. Meanwhile, around 70% of scales are discarded, with less than 30% sold. In addition, local meat prices are three to four times higher than those for scales. Our findings highlight the need to consider entire wildlife trade chains in international policies.

Overexploitation is a major threat to biodiversity globally[1]. At least one-third (~15,000) of vertebrate species are exploited by humans for various products[2], with approximately 24% of wild terrestrial vertebrates traded internationally[3]. Pangolins, eight species of scaly African and Asian mammals, are one such taxon threatened by over-exploitation across their range as well as internationally[4]. Thanks to international demand for their scales, pangolins are among the world's most trafficked wild species[5], despite their inclusion in Appendix I of the Convention on International Trade in Endangered Species of Wild Fauna and Flora (CITES), which bans international commercial trade of wild-caught pangolins[6]. The decline in Asian pangolin populations[7–11] is believed to have triggered large-scale trafficking of African pangolins to supply scales for traditional medicines in parts of Asia[5,12,13]. However, African pangolins have been exploited long before being trafficked to Asia, with their exploitation tied to rural communities' use of wildlife to supplement food and income[14–18].

Although existing research has characterized local dynamics of African pangolin exploitation[15–17,19,20], there is still no explicit assessment

[1]Conservation Science Group, Department of Zoology, University of Cambridge, Cambridge, UK. [2]Wildlife Conservation Society, New York, NY, USA. [3]Pangolin Protection Network, Calabar, Nigeria. [4]Center for Environmental Forensic Science, Department of Biology, University of Washington, Seattle, WA, USA. [5]Center for International Forestry Research and World Agroforestry (CIFOR-ICRAF), Bogor, Indonesia. [6]Interdisciplinary Centre for Conservation Science, University of Oxford, Oxford, UK. [7]Land, Environment, Economics and Policy Institute, Economics Department, University of Exeter, Exeter, UK. [8]Durrell Institute of Conservation and Ecology, School of Natural Sciences, University of Kent, Canterbury, UK. [9]CARE International UK, London, UK. ✉e-mail: emogorcharles@gmail.com

of the proximate drivers of their exploitation—that is, whether hunters are motivated primarily by the international demand for scales, local demand for meat or a combination of both markets and products. Despite this, there is a potentially widespread view among pangolin researchers and stakeholders in Central and West Africa that international demand for pangolin scales is the primary threat to the species in these regions[21].

Here, against the backdrop of intense focus on international trade in pangolin scales, we look at what motivates local stakeholders to kill and trade pangolins. Because these factors may have divergent implications for conservation actions, it is essential to identify the relative importance of scales and meat in driving pangolin exploitation. If pangolin hunting is driven by international demand for scales, conservation efforts could focus on disrupting trade networks, enforcing CITES regulations and running demand-reduction campaigns in consumer countries. Conversely, if local demand for meat is the main driver, prioritizing community engagement through alternative livelihoods or education campaigns may be more effective. In a mixed scenario, where hunters primarily target pangolins for meat but trade scales as a by-product (or vice versa), the most effective interventions are likely those that prioritize addressing the motivations driving demand for the primary product behind their exploitation.

To test the prevalence of these scenarios, we used questionnaires to gather data on the dynamics of pangolin hunting and use from hunters, wild meat market vendors and household members in Nigeria's Cross River Forest landscape[22]. Nigeria is a signatory to CITES and the biggest trade hub for pangolin trafficking globally, with seizures over an 11-year period (2010–2021) involving more than 190,000 kg of scales from an estimated 800,000 African pangolins[5]. Hunting, trading and consuming pangolins are illegal in Nigeria[18,23] but these practices are common, including around the Cross River landscape, a pangolin stronghold and poaching hotspot[21,24]. Black-bellied (*Phataginus tetradactyla*) and white-bellied pangolins (*P. tricuspis*) occur there, with giant pangolins (*Smutsia gigantea*) occurring in adjoining Cameroonian forests. Black-bellied pangolins are classified as vulnerable, while white-bellied and giant pangolins are listed as endangered on the International Union for Conservation of Nature Red List of Threatened Species[4].

Specifically, here we present data on pangolin capture rates, methods, contexts, motivations, and on uses and prices of meat and scales from 590 hunters and 219 wild meat vendors in 33 locations, henceforth called 'hunter and vendor behaviour' data. To validate our responses, we also analyse data on capture rates and meat prices of African brush-tailed porcupine (*Atherurus africanus*), blue duiker (*Philantomba monticola*) and red river hog (*Potamochoerus porcus*), which together represent 57% by mass (56% by number) of the total wildlife offtake in the landscape[25]. Our data span four periods from 2010 to 2023, but we focus here on the most recent (2020–2023, after Nigeria's COVID-19 lockdown) to highlight the current dynamics of pangolin exploitation (see Methods for detail). Additionally, to understand local preferences for different meat types, we present interview data from 190 hunters, 190 vendors and 190 household members across 15 communities in the same landscape to assess the average palatability of pangolin meat and 93 other animal-derived proteins (hereafter 'meat') consumed in our study location, which we refer to as the 'palatability' dataset.

## Results

### Offtake rates

We estimated that between 2020 and 2023, ~21,000 pangolins (confidence interval: 18,200–23,300; 38% black-bellied and 62% white-bellied) were killed annually in the Cross River Forest landscape by an estimated 3,600 hunters (capture rates for other species are presented in Extended Data Fig. 1). The estimated offtake represents approximately 32,700 kg of carcass (meat and scales), with formal hunters

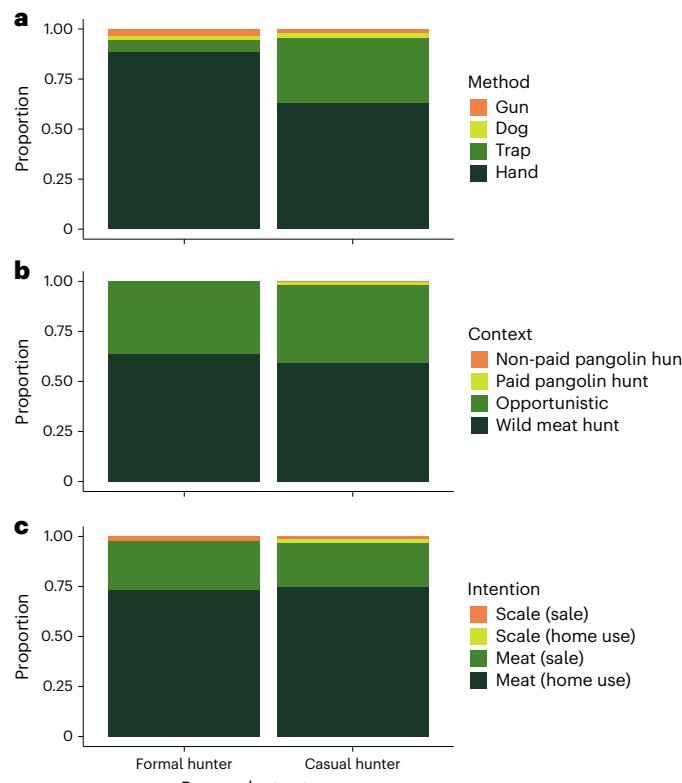

**Fig. 1 | Methods, contexts and intentions associated with pangolin hunting. a**, Pangolins were predominantly captured by hand and trap. **b**, Most pangolin captures occurred during wild meat hunting trips or opportunistically rather than on targeted pangolin hunts. **c**, Hunting pangolins was motivated primarily by demand for their meat for personal consumption and sale. Proportions were calculated using mean values, which gave equal weight to all responses. The panels summarize data from 590 hunters from 32 locations in southeast Nigeria's Cross River Forest landscape.

(who generally hunt using guns) accounting for 59% of pangolin offtake, while casual hunters (who primarily use wire snares) accounted for the remaining fraction. These values make sense given that (a) estimated annual pangolin capture rates across Central African forests are in the hundreds of thousands[15] and (b) an independent three-year dataset derived from direct hunter monitoring in the same landscape also shows a higher capture rate for white-bellied than black-bellied pangolins[25].

### Hunting methods, contexts and intentions

When asked how pangolins were captured, formal hunters told us that simply picking them up by hand was the most common method, with a mean across hunters of 89% (median of 100%) of all formal hunters' captures, compared with 6% by trap, 3% using a dog and 2% using a gun (all responses were given equal weight here and in other calculations). This was slightly different for casual hunters, who caught a mean of 63% (median 70%) by hand, compared with 32% with a wire trap, 3% using a dog and 2% using a gun (median of 20% for wire trap and 0% for other methods; Fig. 1a; results for these and other analyses for periods other than 2020–2023 are presented in Extended Data Figs. 2 and 3).

Our results on the context of hunting revealed that pangolins were rarely caught on dedicated pangolin hunts. Most pangolin captures (mean 64.5%, median 70%) by formal hunters occurred during general hunting trips where hunters targeted most animals they encountered, with a substantial further portion (mean 36%, median 30%) caught opportunistically while doing other, unrelated tasks, such as working in fields. Targeted pangolin hunts (whether pre-financed by a third

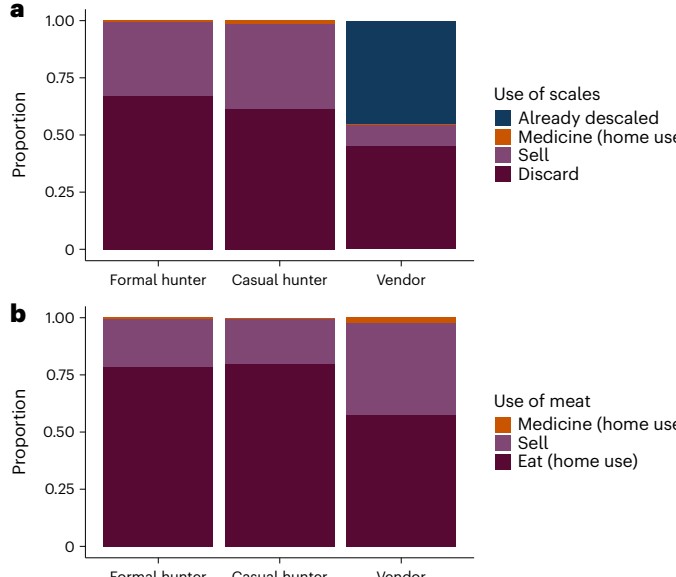

**Fig. 2 | Uses of pangolin scales and meat. a**, Both hunters and vendors discarded a high fraction of pangolin scales without selling them. In nearly half of cases, the animal had already been descaled before being sold to vendors. **b**, Most pangolin meat was eaten at home or sold to be eaten; the use of meat for medicine was negligible. Proportions were calculated using mean values, which gave equal weight to all responses. The panels summarize data from 590 hunters and 219 wild meat vendors from 32 locations in southeast Nigeria's Cross River Forest landscape.

party or not) were virtually non-existent (accounting for a mean of >1% of pangolin captures). Likewise, casual hunters caught a mean of 59% (median 65%) of their pangolins during wild meat hunting trips and 39% (median 30%) opportunistically, with a mean of only 2% caught on dedicated pangolin hunts (median 0%; Fig. 1b).

When asked about their intended use for captured pangolins, formal hunters reported that on most occasions they caught a pangolin in order to consume it as meat (mean across hunters of 73% of occasions; median 80%), to sell pangolin meat on 24% (median 20%) of occasions, and to sell the scales or use them at home just 3% (0%) of the time. Responses were similar from casual hunters, for whom 75% (90%) of pangolin captures were motivated by demand for meat at home with 22% (0%) motivated by selling meat; casual hunters reported their primary intention was to sell or use scales at home on just 3% of occasions (median of 0% for both options; Fig. 1c).

### Uses of pangolin scales and meat
Our results on the uses of pangolin scales showed that both hunters and vendors sold a relatively small fraction of the scales in their possession. The formal hunters we interviewed told us they discarded (that is, threw away) the scales from a mean of 67% (median 100%) of their pangolins, sold 32% and used about 1% for local traditional medicine. Similarly, casual hunters discarded the scales of a mean of 62% (median 100%) of the pangolins they caught and sold 37%, with 1% of captures used for medicinal purposes. Strikingly, even market vendors told us they discarded the scales of a mean of 45% of the pangolins they handled, selling only ~9% and using 1% for personal medicine (with the remainder (45%) of the animals they bought already being descaled; median across all categories for vendors was 0%; Fig. 2a). Turning to our results on uses of meat, all respondent groups told us they ate most of the pangolins they handled. Formal hunters themselves ate a mean of 79% (median 100%) of the pangolins they caught, sold a smaller fraction (20.5%) and used very little for medicine (0.5%). These patterns were similar among casual hunters: mean values of 80% of pangolin captures eaten at home, 19% sold and 1% used for medicine (median

of 100% for eaten at home and 0% for other categories). Vendors also appeared to eat a high fraction (mean 57%, median 60%) of the pangolins they handled and sold 40% (median 20%; Fig. 2b), with 3% used as medicine (median 0%). When probed on the medicinal uses of pangolin derivatives, respondents across the categories told us that pregnant women eat the meat so that they give birth to healthy and strong children, while scales are used as talismans and as ornaments in cultural events.

### Prices of scales and meat, and the palatability of pangolins
A statistical model comparing prices that hunters and vendors charged for the meat and scales from adult black- and white-bellied pangolins over all four of our time periods revealed that, on a per animal basis, a pangolin's meat fetched three to four times as much as its scales. The real prices of both parts of the animals decreased over time, reaching their lowest point during the COVID-19 lockdown period (Fig. 3a; overall $r^2$ = 78%, fixed effects $r^2$ = 45%; full details in Supplementary Table 1). The low price of scales, coupled with an apparently ineffective or nascent supply chain, probably explains why scales were discarded. Temporal trends in the prices of African brush-tailed porcupine, blue duiker and red river hog closely resembled those of pangolins, notably also showing the fall in prices during COVID-19 lockdowns (overall $r^2$ = 90%, fixed effects $r^2$ = 82%; Fig. 3b–d; full details in Supplementary Table 2; note that we fitted two models (one for the pangolin species and the other for the remaining three species) due to multicollinearity in an earlier model with all five species).

Our independent palatability dataset gathered from 570 hunters, vendors and other household members from the Cross River Forest landscape showed that pangolins have the highest palatability compared to all wild and domestic meat, fish or invertebrates that we asked about (Fig. 4). The three pangolin species assessed for palatability all had mean scores across respondents of between 8.73 and 8.90 out of 10 (median of 10 for each pangolin species), possibly explaining why it appears that personal consumption and local sale of pangolin meat drive pangolin hunting. The palatability of pangolins was matched only by that of African brush-tailed porcupines (mean and median of 8.71 and 10, respectively).

## Discussion
The higher mass of scales compared to meat in the international illegal pangolin trade[5,7] may explain why international demand for scales is perceived as the primary driver of African pangolin exploitation[21]. However, our study provides four separate lines of evidence indicating that pangolin hunting in southeast Nigeria may be driven more by domestic demand for their meat. First, captures rarely occurred on dedicated pangolin hunts, with the great majority of pangolins caught opportunistically (35%) or on general hunting trips (62%; Fig. 1b). Second, most pangolins were captured with the primary intention of consuming their meat (74%) or selling the meat locally (23%) rather than for their scales (3%; Fig. 1c). Third, only about one-third of scales were sold, with scales discarded in more than 70% of cases. In contrast, in almost all cases the pangolin meat was consumed (either eaten by the hunter or vendor, or sold locally). Finally, local prices for pangolin scales (on a per animal basis) are extremely low, with the value of the meat from an animal worth roughly three to four times that of its scales.

The data presented in this paper rely on extended recall and self-reported data, which are prone to error and social desirability bias (that is, when people provide responses that they think will make them appear favourably)[26]. Nonetheless, we suggest that our results are relatively robust because: (1) the responses we obtained are corroborated across independent respondent groups within our data; (2) the responses align quantitatively with independently derived data from the same landscape; in particular, our annual offtake rates are comparable to rates observed in a three-year dataset based on direct monitoring of hunters in the same landscape[25]

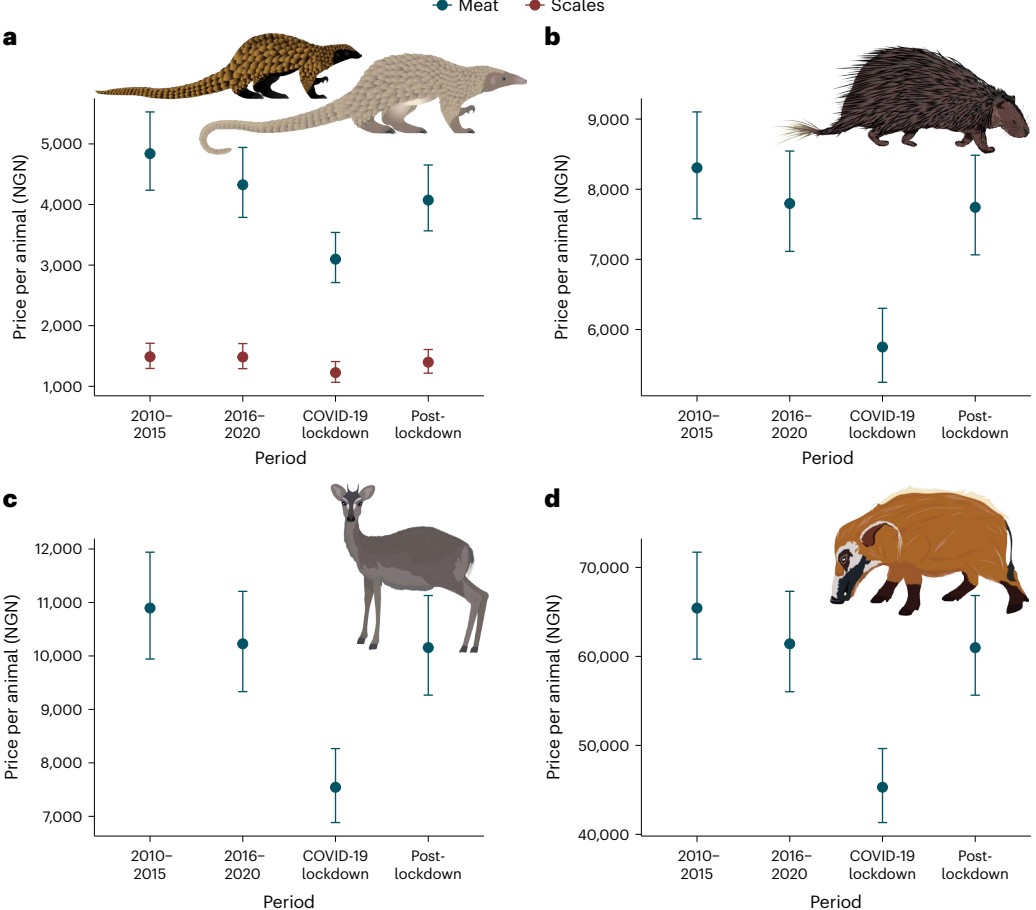

**Fig. 3 | Prices of animal scales and meat. a–d**, Trends in the prices paid to hunters and vendors for pangolin scales and meat, and the meat of three other commonly hunted species. The real price of all the scales from individual adult black- and white-bellied pangolins was lower than that of their meat, with meat prices falling over time more steeply than scale prices; the figure shows combined prices in Nigerian naira (NGN) for both pangolin species (**a**). The trend in pangolin meat price was comparable with that for three other commonly harvested species: African brush-tailed porcupine (**b**), blue duiker (**c**) and red river hog (**d**). The error bars show the effects of the respective variables: the circles are mean predictions, while the vertical lines are 95% confidence intervals. The greenish-blue bars represent meat prices, while the brown bars indicate scale prices. The four time periods are January 2010 to December 2015, January 2016 to February 2020, April to September 2020 (COVID-19 lockdown in Nigeria) and October 2020 to September 2023. The panels summarize data from 528 hunters and 170 wild meat vendors from 32 locations in southeast Nigeria's Cross River Forest landscape (note that some data points were dropped before fitting the models; Methods). Species art by Samudhi Silva and Anupama Dissanayake.

(Supplementary Information and Extended Data Fig. 4); and (3) our respondents freely admitted conducting illegal activities, including killing and trading pangolins, suggesting minimal sensitivity around this topic. Further, our two focal species, averaging around 2 kg in weight, are considerably smaller than other African pangolin species[27,28] (approximately 31 kg for giant pangolins and 10 kg for Temminck's pangolins (*S. temminckii*)[29,30]). As a result, our findings may be specific to black- and white-bellied pangolins (*Phataginus* spp.). However, this does not diminish the importance of our results as *Phataginus* spp. represent approximately 98% of African pangolins trafficked internationally (based on seizure data)[5] and 96% of pangolins caught by hunters across Central and West Africa (based on hunter offtake data from six countries)[15].

### Scales as a by-product of hunting for meat

Our findings indicate that pangolin-specific hunting for scales contributes negligibly to the overall exploitation of these species in southeast Nigeria, specifically around our study location (Extended Data Fig. 5). Rather, it seems that those scales which are trafficked from the landscape are a by-product of pangolins captured for their meat for local consumption. Note that the consumption of pangolin meat across sub-Saharan Africa, and certainly in our study location, is different from many rural Asian communities that are increasingly selling the commodity to cities where eating meat signals high social status[31] and is therefore a luxurious activity[8,11]. Further, the higher price of pangolin meat than scales in our study indicates that meat may be in greater demand, with this demand probably driven by pangolin meat's exceptional palatability. Taken together, it appears that successfully ending pangolin trafficking without addressing local exploitation for meat will have minimal effect on the species' survival. Decisions to trade in pangolin scales are potentially influenced by people's awareness of their value and presence of supply chains[32]. While we did not directly investigate the prevalence of these factors, it is conceivable that there is local knowledge of supply chains, as demand for scales occasionally drove hunting, although negligibly (Fig. 1c). That said, our finding that scale prices have been lower than meat since at least 2010 suggests the absence of a premium on scales locally, with only a modest market demand for the commodity.

It is possible that there is a bespoke supply chain for scales in southeast Nigeria, involving hunters and buyers solely focused on pangolin scales, which we may not have detected. However, we consider it improbable that there are specialized pangolin hunters operating in the region. If there were, it is likely that some of the hunters, households and vendors in our study would have known and informed us; also, local

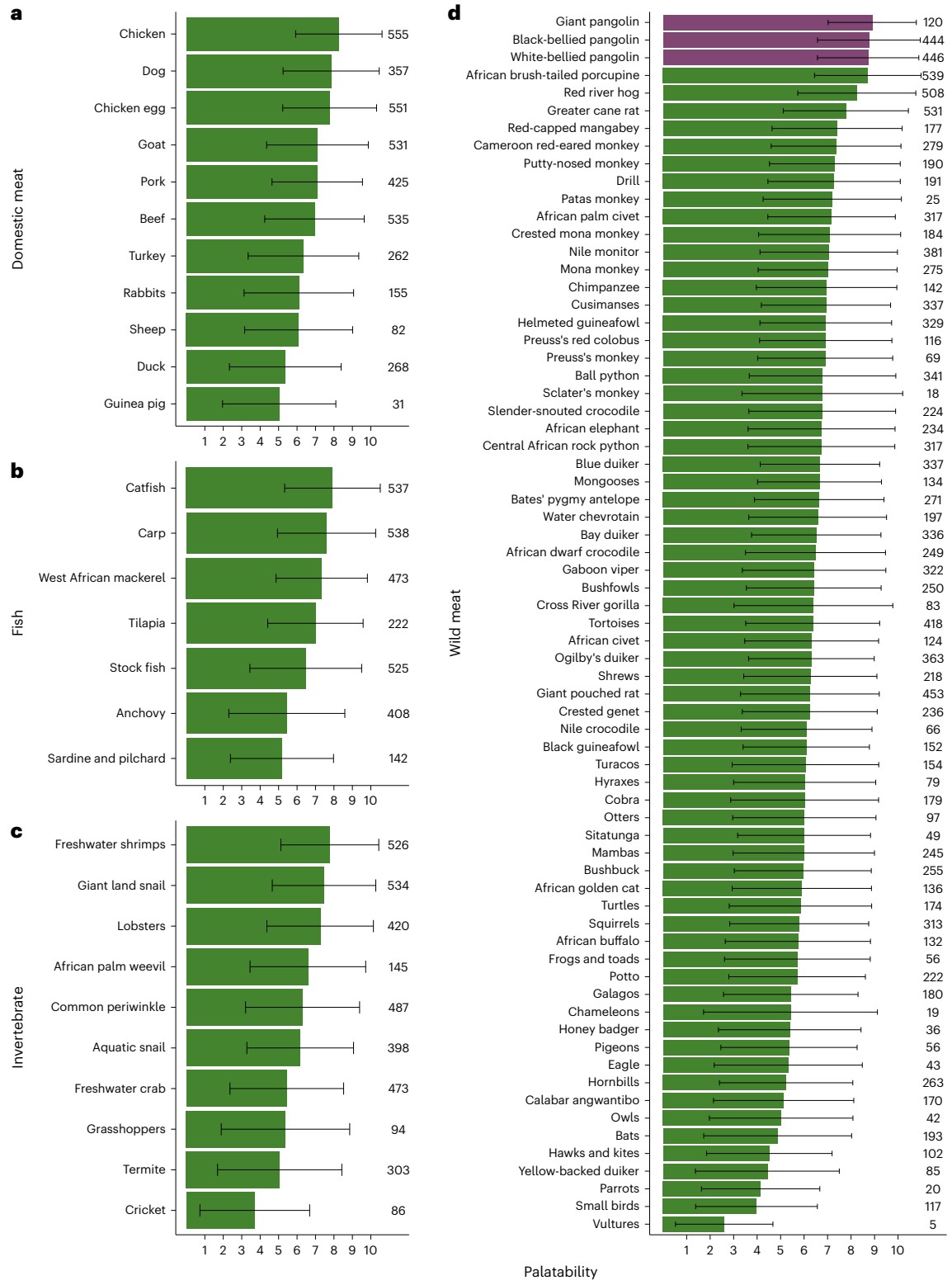

**Fig. 4 | The high palatability of pangolin meat. a–d**, Pangolin meat (shown in purple) was more palatable than all assessed domestic meat (**a**), fish (**b**), invertebrate (**c**) and wild meat (**d**), except African brush-tailed porcupine. Note that the charts are ordered from highest to lowest average palatability (top to bottom). The figure summarizes data from 570 hunters, vendors and household members from the same landscape. Error bars are 95% confidence intervals (derived by calculating the standard deviation of the scores per species), and the right-hand values give the number of respondents who scored each meat.

prices for scales would have been higher than we found. Nonetheless, we do not completely dismiss the possibility of such a bespoke supply chain, and therefore acknowledge that our findings show only the prevalence of a largely overlooked driver of pangolin exploitation

among general hunters and vendors represented in our study. We also appreciate that, although scales may currently be a by-product of pangolin hunting in southeast Nigeria, in the long-term—especially without appropriate conservation actions—a more established market

for the commodity could emerge, which may incentivize hunters to target pangolins to meet overseas demand for scales.

## Looking beyond Nigeria

While our findings may not apply across all African pangolin ranges, there are at least three reasons why these results could hold in other forest landscapes in Central and West Africa where pangolins are eaten. First, the proportion of pangolins in the overall hunter offtake in our study landscape (approximately 2%)[25] is similar to figures reported across these regions, based on data from sites in Cameroon, Central African Republic, Democratic Republic of Congo, Equatorial Guinea, Gabon and the Republic of Congo[15]. Second, in addition to being consumed for food in Central and West Africa[33], pangolins are also considered highly palatable in Cameroon[34], Equatorial Guinea[35], Gabon[36] and the Republic of Congo[17]. Last, pangolin scales in our study were traded at approximately US$13 per kg, similar to prices reported in the Republic of Congo[17], Cameroon and Uganda for similar periods[37]. In contrast, scales fetch around US$500–1,000 per kg in parts of Asia[38,39].

## Implications for pangolin conservation

Our findings of the importance of meat in motivating pangolin hunting offers valuable insights for the conservation of these species, whose exploitation is already considered unsustainable[5,15]. First, other things being equal, the cost of implementing site-based interventions for pangolin exploitation driven by local meat demand would probably be lower than where it is motivated by high prices for scales. Second, since most pangolin scales are discarded, relying on data on the trade in scales to assess the species' conservation status is likely to vastly underestimate pangolin exploitation. Third, law enforcement efforts targeting traffickers are unlikely to reduce pangolin hunting substantially in this part of Africa. Instead, priority should be given to site-level interventions, such as anti-poaching patrols and community-based actions, including initiatives to improve food security and behaviour change programmes for hunters. Although patrols can deter poaching and reduce threats to pangolins via snare removals[40], they are mostly confined to protected areas and are often limited in their effectiveness because of insufficient resources to ensure adequate patrol effort[41,42]. Behaviour change interventions combined with food security programmes may be effective and well-received by local communities, as they address direct needs without antagonizing them[18,43]. However, caution is needed during project design to ensure compliance and that hunting does not shift to other threatened species[44]. Maintaining local support is also critically important to reduce the risk of increased pangolin hunting should the market for Nigerian-derived scales increase.

Looking beyond pangolins, our results underscore the importance of incorporating considerations of domestic drivers of species exploitation into decision-making within international wildlife trade treaties[45]. Among taxa with biological resource use as a threat, four times more species are threatened by local use than international trade[46]. Our results thus highlight an important pathway where international trade regulations may not reduce exploitation pressures, as supplying international trade may not be the primary reason species are hunted. Therefore, focusing on trade restrictions without complementary local measures to curb exploitation in and around species habitats may prove ineffective[45,47]. This consideration is especially crucial for CITES Appendix II species, whose commercial trade is permitted if the trade does not harm the species' survival in the wild. CITES and other stakeholders could support countries to assess domestic drivers of exploitation of listed species, particularly those consumed locally as food (that is, wild meat)[48,49]. Where these drivers differ from those of international trade, tailored interventions should be designed and implemented to simultaneously address domestic exploitation.

## Methods

### Study location

We worked in southeast Nigeria's Cross River Forest landscape, which lies in a global biodiversity hotspot[50], contains the largest block of forest in Nigeria and includes three protected areas: Afi Mountain Wildlife Sanctuary (100 km²), Mbe Mountains Community Forest (86 km²) and Cross River National Park (CRNP; 4,000 km², comprising Oban and Okwangwo divisions; Extended Data Fig. 5). CRNP is contiguous with Cameroon's Takamanda and Korup national parks. Hunting for meat is the primary threat to large-bodied vertebrate species in the landscape[51]. Hunting for certain taxa (such as primates and pangolins) is prohibited both within and outside protected areas but permitted for others (certain rodents) when caught outside park boundaries[52]. We received ethics approval for this study from Cambridge University's Psychology Research Ethics Committee (applications: PRE.2023.097; the information sheet for participants and consent form are in the Supplementary Information). Study participants provided written, free and informed consent before we commenced the survey. We anonymized all data and mostly interviewed adult volunteers (from 18 years of age), with parents or guardians providing consent where volunteers were below 18.

### Data collection

This section outlines the data collection protocols for our two surveys. The main survey focused on hunters and vendors (809 respondents across 33 locations; hunter and vendor behaviour survey), while the second survey involved hunters, vendors and household members (570 respondents across 15 locations; palatability survey). We conducted the surveys a year apart, but the datasets are not entirely independent, as 11 locations targeted in the hunter and vendor behaviour survey were also used for the palatability survey (33% overlap).

**Hunter and vendor behaviour survey.** In collecting the primary dataset, we deployed structured questionnaires (October to November 2023) to survey 590 hunters (392 formal and 198 casual hunters; 99% men and 1% women) in 20 rural communities, and 219 wild meat vendors across the communities and 13 other locations (including four towns or cities; 65% women and 35% men). To select the 20 focal communities out of 144 (14% of the total), we combined stratified random and purposive sampling. Our random sampling involved dividing the two CRNP divisions into four geographic quadrants (hereafter strata; our stratification captured other protected areas in the landscape) and randomly selecting 12 communities (two to four per stratum based on their total number of communities, except in southeast Okwangwo where no community occurs). We then purposively added a further eight communities to the selection—these communities represent locations wherein we had ongoing wild meat exploitation research.

Next, we obtained the permission of community leaders and started data collection by counting all households in each community (household census), defining a household as a group of people living under the same roof and sharing the same meals. We aimed to sample all hunters and vendors in each community, so during the count we asked if a household member was a hunter or vendor. We later returned to households with hunters and vendors to administer the questionnaire (some hunters and vendors were absent at the time of interview, while 19 people declined to take part). We estimate that our final survey sample represents 20% of hunters and 43% of vendors across the entire landscape. We supplemented the data from vendors by visiting wild meat markets in these communities.

We structured the questionnaire into five interrelated sections: (a) pangolin capture rate; (b) pangolin capture methods; (c) hunting context; (d) intentions when catching pangolins; (e) uses of pangolin derivatives (meat and scales); and (f) prices of pangolin derivatives

(the questionnaire for vendors contained only sections e and f). To track trends over time, we asked the questions in sections b–f for four different periods, beginning in 2010 as Nigeria's first pangolin scale seizure was documented then[5]: (i) January 2010 to December 2015; (ii) January 2016 to February 2020; (iii) April to September 2020 (COVID-19 lockdown in Nigeria); and (iv) October 2020 to September 2023 (our census and hunter and vendor behaviour questionnaires are provided in Supplementary Table 3). Given the similarity in responses across the periods, our results focus on the most recent period (results on other periods are in Extended Data Figs. 2 and 3).

To estimate pangolin capture rates, we asked about the average number of pangolins which hunters had killed in the wet (April to October) and dry (November to March) seasons over the past three years. Sections b–e involved asking respondents to opt for different answers in a proportional fashion, following the weighted ranking method[53]. In this process, we provided ten pebbles and asked the participants to distribute the pebbles among the options, with the number of pebbles allocated to each option indicating its contribution to their answer. We then converted these values to proportions for each question using both median and mean averages (giving equal weights to all responses). Regarding the methods used, we asked whether pangolins were hunted using (1) gun, (2) dog, (3) trap (for example, wire snare) and (4) hand. To understand the context of hunting, we inquired whether pangolins were hunted (1) during wild meat hunt, (2) opportunistically (that is, when not hunting), (3) on paid pangolin hunts (that is, commissioned by a third party) or (4) on non-paid pangolin hunts. To understand motivations, we asked hunters whether they hunted pangolins to obtain (1) meat for home use, (2) meat for sale, (3) scales for sale or (4) scales for home use. The options on possible uses of pangolin meat were (1) eat at home, (2) use for medicine at home, (3) sell and (4) discard; and those on the uses of scales were (1) sell, (2) use for medicine at home, (3) discard and (4) no access to scales (for those cases where vendors purchase animals that have already been descaled). On prices, we asked about the average price in each period of meat from adult black- and white-bellied pangolins and the complete scales of adult individuals, as well as prices for African brush-tailed porcupine, blue duiker and red river hog, which together comprise about half of the total offtake (by number of animals) in the landscape[25]. The interviews lasted an average of 40 min for hunters and 20 min for vendors.

**Palatability survey.** To compare the palatability of pangolin meat with other meats, we used data collected in August to September 2022 from 190 hunters, 190 vendors and adult members of 190 households across 15 communities in the Cross River Forest landscape. In addition to black-bellied, white-bellied and giant pangolins, we also asked about the palatability of 93 other types of solid animal protein (which, for simplicity, we term 'meat'). Together they represent the main domestic meat and eggs, fish, invertebrates and wild meat consumed as food by humans in the landscape. Data collection involved assigning scores ranging from 1 to 10 to each meat respondents reported eating, with 10 representing the highest palatability. Note that giant pangolins are possibly extinct in CRNP but still occur in contiguous forests in Cameroon. The questionnaire and median scores per meat are provided in Supplementary Table 4, with a detailed data collection protocol described in ref. 54.

### Quantifying pangolin extraction

To estimate landscape-level pangolin extraction, we first derived the mean and 95% confidence intervals of the number of animals reportedly killed annually (per species and for each hunter category) using 1,000 bootstrapped replicates (R boots package[55]). We used the adjusted bootstrap percentile method to compute confidence intervals to account for skewness in the data[56]. Note that we used

the mean, despite skewness in the data, as it was not possible to calculate confidence intervals for some species using the median due to the small variance in the data. Next, we multiplied these values by the median number of hunters in each community (per hunter category), which we derived through our household census ('Data collection'). We then multiplied these estimated totals per community by the total number of communities in the landscape; Supplementary Table 5)—assuming that the number of hunters in our focal communities (obtained via the census) is representative of that of other communities in the landscapes. We obtained extraction rates for the other species following these same steps. For pangolins we converted these resulting values to carcass mass by multiplying the estimated number of pangolins killed annually by the median dried mass of meat and complete scales for each pangolin species (Supplementary Table 6).

### Temporal trend in prices

To understand changes in the price of pangolin scales and meat over time, we fitted a mixed effects model with our response variable being each respondent's assigned price per part (meat or scales) for each of the four periods ('Data collection'). We used data from 431 respondents because we dropped those who did not provide prices for either meat or scales for all four periods (note also that we discarded 80 records where scale prices were incorrectly collected on a per kilogram instead of per animal basis). Our predictor variables were respondent type (vendor or hunter), period and part (all categorical variables), with an interaction between period and part to explore whether prices changed differently for meat and scales. As we had multiple responses per respondent and, in some cases, more than one respondent per household, we used a nested random effect model (that is, nesting respondents within unique households). Furthermore, we specified a random effect of community to control for possible similarities in responses from respondents in the same community.

We adjusted the nominal prices for each period to reflect current prices (that is, real prices in 2023) by multiplying the values for each period by the inflation rate in 2023. We derived inflation rates using Nigeria's consumer price index[57,58], thus using median consumer price index per period.

To examine whether the temporal trends in prices for pangolin parts were similar to those of other species, we fitted a similar mixed effects model of each respondent's assigned price per period for African brush-tailed porcupine, blue duiker and red river hog as a function of period and respondent type ($n = 690$ respondents who provided prices per species for all periods). As in the pangolin price model, we nested each respondent within households and included a separate random effect of community. We dropped an initially specified interaction between species and period due to collinearity (high variance inflation factor values after fitting the model).

Both models were fitted with data from a total of 698 respondents (comprising 11,876 data points; 528 hunters and 170 vendors); 424 respondents provided data for all species (and pangolin parts) across the four periods. All analyses were conducted using R v.4.2.2 (ref. 59), with lme4 (ref. 60) and emmeans[61] packages used to fit the models and conduct post hoc tests, respectively. We assessed model fit using the R performance package[62] (distribution of raw data and model fit are presented in Supplementary Figs. 1–4).

### Reporting summary

Further information on research design is available in the Nature Portfolio Reporting Summary linked to this article.

### Data availability

The data used in this study are available via Zenodo at https://doi.org/10.5281/zenodo.15084096 (ref. 63).

## Code availability

All analyses were conducted using the R statistical environment (v.4.2.242) within the open-access integrated development environment, RStudio. We used the lme4 package (v.1.1.36) to fit the models, emmeans (v.1.11.0) to perform post-hoc tests and performance (v.0.13.0) to assess model fit. Data processing and visualization were carried out using tidyverse (v.2.0.0), ggpubr (v.0.6.0), lubridate (v.1.9.4) and ggsci (v.3.2.0). To estimate capture rates through bootstrapping, we used the boot package (v.1.3.31). The study map was created using QGIS (v.3.42). The codes used to develop the models are available via GitHub at https://github.com/cemogor/pangolin-exploitation-in-southeast-nigeria.

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

## Acknowledgements

We thank community leaders for permission to gather data and the study hunters, vendors and household members for participating in our study. We also thank D. O. Agbor and N. M. Effa for supporting data collection. Funding for data collection was provided by the Pangolin Protection Network through the Wildlife Conservation Society's Local Conservation Partners Fund, established with a grant from Arcadia, a charitable fund of L. Rausing and P. Baldwin. We acknowledge funding from the Gates Foundation (grant no. OPP1144) and Schmidt Science Fellows (C.A.E.), UK Research and Innovation (Future Leaders Fellowship, grant no. MR/W006316/1; D.J.I.), Dragon Capital Chair on Biodiversity Economics (B.B.), UK Research and Innovation's Global Challenges Research Fund through the Trade, Development and the Environment Hub project (grant no. ES/S008160/1; L.C.) and United States Agency for International Development to CIFOR (C.A.E. and L.C.). Article processing charge was covered by the University of Cambridge. The extended research credits in the Supplementary information contains detailed credit.

## Author contributions

C.A.E. and A.B. conceptualized the study and carried out project administration. C.A.E., D.J.I., A.W., L.C. and A.B. designed the methodology. C.A.E. undertook the investigation, funding acquisition and wrote the initial draft. C.A.E., B.B. and D.J.I. were responsible for visualization. A.B. supervised the study. S.K.W., L.C., B.B., D.J.I., A.W. and A.B. reviewed and edited the paper.

## Competing interests

C.A.E. is the founder of Pangolin Protection Network (aka Pangolino; https://pangolino.org/), a conservation non-profit promoting community-based interventions to reduce pangolin decline in Nigeria. A.B. acts as Pangolino's scientific adviser. All other authors declare no competing interests. Most authors are either Pangolino staff or volunteers (see affiliations).

## Additional information

**Extended data** is available for this paper at https://doi.org/10.1038/s41559-025-02734-3.

**Correspondence and requests for materials** should be addressed to Charles A. Emogor.

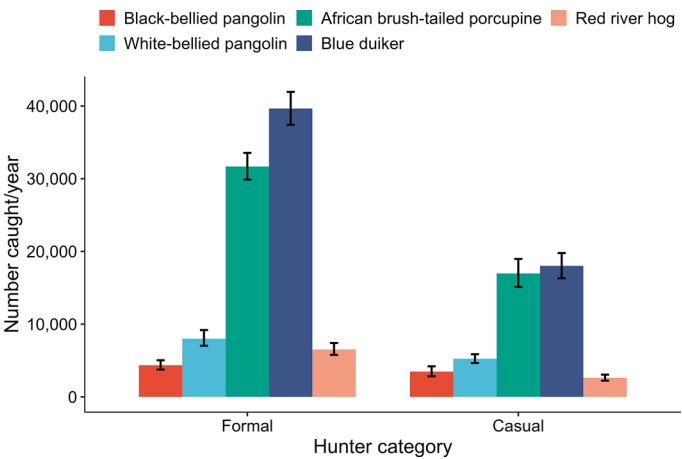

**Extended Data Fig. 1 | The estimated number of animals per species killed annually across the communities in the Cross River Forest landscape (n = 144).** Error bars are 95% confidence intervals derived using 1000 bootstrapped replicates of the reported number of individuals of each species killed annually.

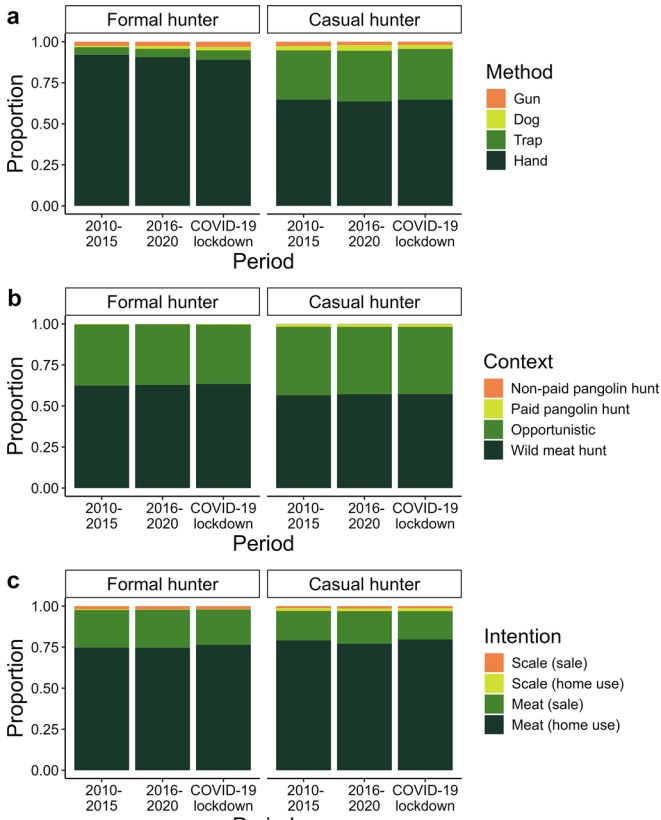

**Extended Data Fig. 2 | Consistency across periods in the methods (a), contexts (b), and intentions (c) associated with hunting pangolin from 2010 to 2020 in southeast Nigeria's Cross River Forest landscape.** Proportions were calculated using mean values, which gave equal weight to all responses. The exact duration of the periods, in the order presented, are January 2010-December 2015, January 2016-February 2020, and April-September 2020 (COVID-19 lockdown in Nigeria). Data for September 2020 – September 2023 are presented in the main text. The panels summarise data from 590 hunters.

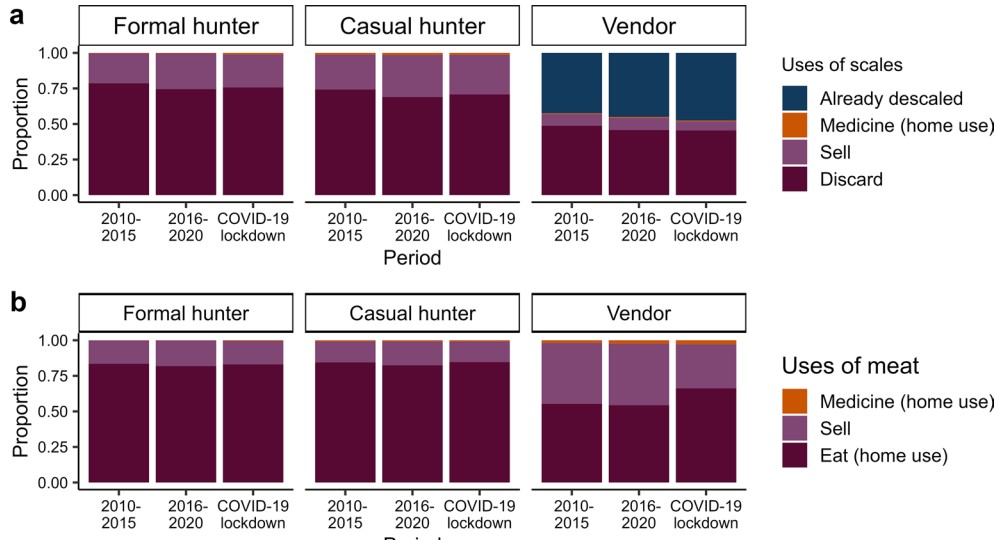

**Extended Data Fig. 3 | Consistency across periods in the reported uses of pangolin scales (a) and meat (b) from 2010 to 2020 in southeast Nigeria's Cross River Forest landscape.** Proportions were calculated using mean values, which gave equal weight to all responses. The exact duration of the periods, in the order presented, are January 2010-December 2015, January 2016-February 2020, and April-September 2020 (COVID-19 lockdown in Nigeria). Data for September 2020 – September 2023 are presented in the main text. The panels summarise data from 590 hunters and 219 wild meat vendors.

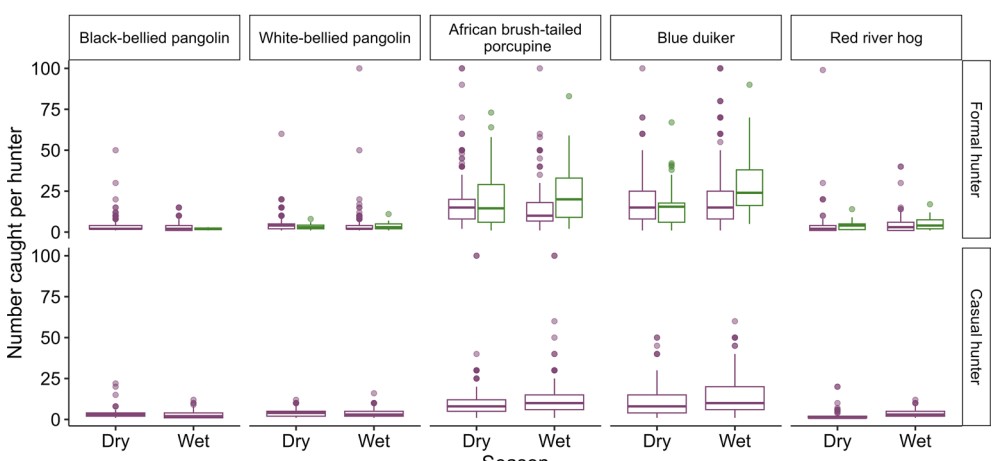

**Extended Data Fig. 4 | The number of individuals of the five species that formal and casual hunters report they, on average, kill in each season (labelled as recall) and the number observed through three years of monitoring hunting activities of 33 formal hunters in the same landscape (direct monitoring)25.** The wet season spans April to October, while the dry season runs from November to March. The number of individuals of each species caught is shown per season for the five species indicated at the top of each panel. The category of hunter is shown on the horizontal panel header. The thick horizontal bar in each boxplot (or thick vertical bar) shows the median, the box indicates the interquartile range, and the lines show the overall range, with outliers marked by dots.

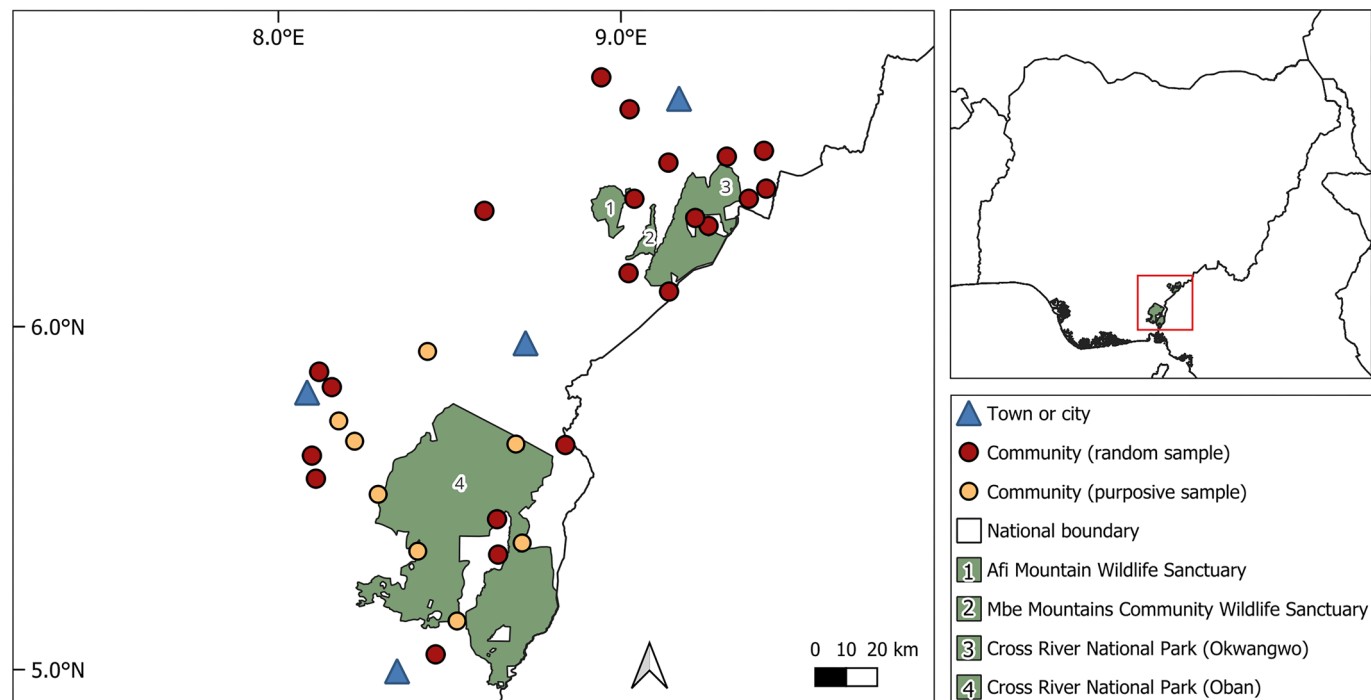

**Extended Data Fig. 5 | Cross River Forest landscape showing the approximate locations where we interviewed respondents.** The top left map shows the landscape location in Nigeria. Note that we have withheld community names to ensure their anonymity, in line with our ethics approval requirement. Shapefiles of Nigeria and Africa were downloaded via openAfrica (https://open.africa/) while protected area shapefiles were provided by the Wildlife Conservation Society, Nigeria.

# Reporting Summary

## Statistics

For all statistical analyses, confirm that the following items are present in the figure legend, table legend, main text, or Methods section.

| n/a | Confirmed | |
|---|---|---|
| ☐ | ☒ | The exact sample size (*n*) for each experimental group/condition, given as a discrete number and unit of measurement |
| ☐ | ☒ | A statement on whether measurements were taken from distinct samples or whether the same sample was measured repeatedly |
| ☐ | ☒ | The statistical test(s) used AND whether they are one- or two-sided<br>*Only common tests should be described solely by name; describe more complex techniques in the Methods section.* |
| ☐ | ☒ | A description of all covariates tested |
| ☐ | ☒ | A description of any assumptions or corrections, such as tests of normality and adjustment for multiple comparisons |
| ☐ | ☒ | A full description of the statistical parameters including central tendency (e.g. means) or other basic estimates (e.g. regression coefficient) AND variation (e.g. standard deviation) or associated estimates of uncertainty (e.g. confidence intervals) |
| ☐ | ☒ | For null hypothesis testing, the test statistic (e.g. *F*, *t*, *r*) with confidence intervals, effect sizes, degrees of freedom and *P* value noted<br>*Give P values as exact values whenever suitable.* |
| ☒ | ☐ | For Bayesian analysis, information on the choice of priors and Markov chain Monte Carlo settings |
| ☒ | ☐ | For hierarchical and complex designs, identification of the appropriate level for tests and full reporting of outcomes |
| ☒ | ☐ | Estimates of effect sizes (e.g. Cohen's *d*, Pearson's *r*), indicating how they were calculated |

*Our web collection on statistics for biologists contains articles on many of the points above.*

## Software and code

Policy information about availability of computer code

Data collection | No proprietary software or custom code was used for data collection in this study. The code used for regression model fitting is available on GitHub at https://github.com/cemogor/pangolin-exploitation-in-southeast-nigeria.

Data analysis | All analyses were conducted using the R statistical environment (v.4.2.242) within the open-access integrated development environment, RStudio (https://posit.co/download/rstudio-desktop/). We used the lme4 package (v.1.1.36) to fit the models, emmeans (v.1.11.0) to perform post-hoc tests, and performance (v.0.13.0) to assess model fit. Data processing and visualization were carried out using tidyverse (v.2.0.0), ggpubr (v.0.6.0), lubridate (v.1.9.4) and ggsci (v.3.2.0). To estimate capture rates through bootstrapping, we used the boot package (v.1.3.31). The study map was created using QGIS (v.3.42).

For manuscripts utilizing custom algorithms or software that are central to the research but not yet described in published literature, software must be made available to editors and reviewers. We strongly encourage code deposition in a community repository (e.g. GitHub). See the Nature Portfolio guidelines for submitting code & software for further information.

# Data

Policy information about availability of data

All manuscripts must include a data availability statement. This statement should provide the following information, where applicable:

- Accession codes, unique identifiers, or web links for publicly available datasets
- A description of any restrictions on data availability
- For clinical datasets or third party data, please ensure that the statement adheres to our policy

The data used in this study are available on Zenodo via this link: https://doi.org/10.5281/zenodo.15084096

# Research involving human participants, their data, or biological material

Policy information about studies with human participants or human data. See also policy information about sex, gender (identity/presentation), and sexual orientation and race, ethnicity and racism.

| | |
|---|---|
| Reporting on sex and gender | While our study relies on data from human participants, we did not include sex or gender as covariates in any of our models. |
| Reporting on race, ethnicity, or other socially relevant groupings | NA |
| Population characteristics | All research participants were either hunters, wild meat vendors, or other household members living in southeast Nigeria. |
| Recruitment | Our recruitment was systematic. To select 20 focal communities from 144 (14% of the total), we divided the two Cross River National Park divisions into four geographic quadrants (strata), including other protected areas. We randomly selected 12 communities (2-4 per stratum, except one strata where no community exists) and purposely sampled an additional eight communities where we were conducting research. Hunters were recruited at their homes, and vendors were recruited both at their homes and in wild meat markets. |
| Ethics oversight | We received ethics approval for this study from Cambridge University's Psychology Research Ethics Committee (application number: PRE.2023.097). Study participants provided written, free, and informed consent before we commenced the survey and all data were anonymised. |

Note that full information on the approval of the study protocol must also be provided in the manuscript.

# Field-specific reporting

Please select the one below that is the best fit for your research. If you are not sure, read the appropriate sections before making your selection.

☐ Life sciences   ☒ Behavioural & social sciences   ☐ Ecological, evolutionary & environmental sciences

For a reference copy of the document with all sections, see [nature.com/documents/nr-reporting-summary-flat.pdf](http://nature.com/documents/nr-reporting-summary-flat.pdf)

# Behavioural & social sciences study design

All studies must disclose on these points even when the disclosure is negative.

| | |
|---|---|
| Study description | We used a mixture of self-reported qualitative and quantitative data for this study, some of which we verified using an independent dataset from the same region. |
| Research sample | The data used in this study were collected in 2023. The core data came from 809 local hunters and wild meat vendors in southeast Nigeria, with supplementary data on meat palatability from 570 hunters, vendors, and households in the same landscape. Cross River forest landscape, in southeast Nigeria, is one of the largest remaining forest blocks in the Guinean Forest biodiversity hotspot. The landscape is a hotspot for pangolin scale exports, as well as a stronghold for African pangolins in Africa. Nigeria is a major hub in the global, illegal pangolin trade. |
| Sampling strategy | Our recruitment was systematic. To select 20 focal communities from 144 (14% of the total), we divided the two Cross River National Park divisions into four geographic quadrants (strata), including other protected areas. We randomly selected 12 communities (2-4 per stratum, except one strata where no community exists) and purposely sampled an additional eight communities where we were conducting research - some of the additional communities were randomly selected. Hunters were recruited at their homes, and vendors were recruited both at their homes and in wild meat markets.<br><br>After informing community leaders of our study and requesting their permission to conduct the surveys, we counted all households in each community, defining a household as a group of people living under the same roof and sharing the same meals. We aimed to sample all hunters and vendors in each community, so during the count we asked if a household member was a hunter or vendor. We then returned to households with our respondents of interest to go through the questionnaire with them individually (note that some hunters and vendors declined to take part, and others were absent). We then recruited additional vendors through market |

| | |
|---|---|
| | visits.<br><br>We followed a similar approach in recruiting respondents for the palatability survey. |
| Data collection | Data were collected using a standardized survey on Kobotoolbox (https://www.kobotoolbox.org/) via Samsung tablets. Interviews were held with individuals not in groups. |
| Timing | The core data were collected in October to November 2023, with palatability data gathered in August-September 2022. |
| Data exclusions | In our linear model of price of animal part over time, of the 809 respondents, we used data from 431 respondents, as we dropped those who did not provide prices for either meat or scales for all four periods: i) January 2010-December 2015; ii) January 2016-February 2020; iii) April-September 2020 (COVID-19 lockdown in Nigeria); and iv) October 2020-September 2023. Note also that we already discarded 80 records where scale prices were incorrectly collected on a per kilogram instead of per animal basis. |
| Non-participation | No participant dropped from the study. However, 19 people declined to take part. |
| Randomization | We randomly selected focal communities after overlaying four strata in each of the two divisions of the Cross River National Park, the largest protected area in the landscape. Further, we applied systematic approach in gathering data from participants (except the vendors recruited from wild meat markets). |

# Reporting for specific materials, systems and methods

We require information from authors about some types of materials, experimental systems and methods used in many studies. Here, indicate whether each material, system or method listed is relevant to your study. If you are not sure if a list item applies to your research, read the appropriate section before selecting a response.

## Materials & experimental systems

| n/a | Involved in the study |
|---|---|
| ☒ | ☐ Antibodies |
| ☒ | ☐ Eukaryotic cell lines |
| ☒ | ☐ Palaeontology and archaeology |
| ☒ | ☐ Animals and other organisms |
| ☒ | ☐ Clinical data |
| ☒ | ☐ Dual use research of concern |
| ☒ | ☐ Plants |

## Methods

| n/a | Involved in the study |
|---|---|
| ☒ | ☐ ChIP-seq |
| ☒ | ☐ Flow cytometry |
| ☒ | ☐ MRI-based neuroimaging |

## Plants

| | |
|---|---|
| Seed stocks | NA |
| Novel plant genotypes | NA |
| Authentication | NA |

