## [Peer Review File · Nature Ecology & Evolution]

Pangolin hunting in Southeast Nigeria motivated more by local meat consumption than international demand for scales

Corresponding Author: Dr Charles Emogor

Version 0:

Decision Letter:

20th October 2024

Dear Dr Emogor,

Your manuscript entitled "Pangolin hunting in Nigeria motivated mainly by local meat consumption not international demand for scales" has now been seen by three reviewers, whose comments are attached. The reviewers have raised a number of concerns which will need to be addressed before we can offer publication in Nature Ecology & Evolution. We will therefore need to see your responses to the criticisms raised and to some editorial concerns, along with a revised manuscript, before we can reach a final decision regarding publication.

We therefore invite you to revise your manuscript taking into account all reviewer and editor comments. Please highlight all changes in the manuscript text file.

* If you have not done so already please begin to revise your manuscript so that it conforms to our Article format instructions at <http://www.nature.com/natecolevol/info/final-submission>. Refer also to any guidelines provided in this letter.

Link Redacted

Nature Ecology & Evolution is committed to improving transparency in authorship. As part of our efforts in this direction, we are now requesting that all authors identified as 'corresponding author' on published papers create and link their Open Researcher and Contributor Identifier (ORCID) with their account on the Manuscript Tracking System (MTS), prior to acceptance. ORCID helps the scientific community achieve unambiguous attribution of all scholarly contributions. You can create and link your ORCID from the home page of the MTS by clicking on 'Modify my Springer Nature account'. For more

information please visit please visit www.springernature.com/orcid.

[redacted]

Reviewer expertise:

Reviewer #1: wildlife trade, pangolins

Reviewer #2: sustainable use of wild resources, wildlife trade, local livelihoods

Reviewer #3: conservation conflicts, wild meat hunting, survey methods

Reviewer comments:

Reviewer #1 (Remarks to the Author):

The authors present important data from hunters and vendors in Nigeria on the trade of pangolins. Indeed, as a major hub in the international trafficking of pangolins, such data is important for context in the conservation of these important species. In general, I feel that the manuscript is well written, the methods are robust, and the results quite clear.

However, I feel that there is a strawman argument presented that I don't find convincing. The suggestion that 'pangolin researchers believe that international demand for scales is a primary driver of pangolin exploitation' just doesn't ring true to me. I'm a pangolin researcher and this has never been my belief. It simplifies what I believe to be a fairly complex issue.

As stated clearly in the paper, African pangolins have been hunted for local community use for, presumably, as long as those communities have been there. We know from the data that the pangolins are and have been hunted for their meat. If, as suggested by the paper, the scales are not important in driving hunting here then that could mean a few things as I see it: 1) these pangolin populations haven't changed much and they're not in fact overexploited... they're hunted for their meat as they have been for centuries, 2) even though scales are not important, hunting has increased recently, perhaps because human population sizes have increased. Otherwise, I think it's important to consider another option consistent with the results here: 3) the scales are in fact important – while hunting for scales is not the “primary” reason to hunt pangolins it still may provide sufficient incentive to increase hunting rates into a state of overexploitation.

My issue with the paper is that the data doesn't provide evidence for or against either of these possibilities, but the primary argument of the paper is against option 3. But I don't think that showing 1) the price of scales is lower than meat, nor 2) most hunters hunt for the meat, necessarily negate this possibility. From Fig 1, we know that the price of scales is ~1500 NGN while meat is ~5000 NGN. That's still presumably decent money? This is evident from the fact that 1/3 of the scales are sold. In the paper this is presented as “only about one-third of scales were sold”. If 20,000 pangolins are hunted, and the scales from 6,000 of those enter the market, this strikes me as potentially concerning. It could be sufficiently added incentive, on top of the primary motivation to hunt them for meat, to lead to overexploitation. I don't know if this is the case or not... but to use the data here to suggest that this is NOT the case appears flawed to me.

Simo et al. *Oryx* 2023 has similar data which I find instructive for comparison. Prices for scales, for white and black bellied, reported in that study are similar to this one but the scales of giant pangolin were more than “15 times the value estimated for the scales of white-bellied pangolin”. So, it's worth noting that there are likely some species-specific differences here with respect to meat vs scale prices. Also, while the price results in that study and this one are similar, Simo et al. found evidence indeed of bespoke supply chains for scales. My best guess behind this difference is the fact there is likely to be strong regional variation in these sorts of things (but this undermines the “Looking beyond Nigeria” result to some extent).

Of course, I agree that we need more attention on what's happening “on the ground” with hunters/vendors and, as noted earlier, I do find great value in the data collected. But I don't think there is good evidence presented to support the idea that scales are not important and that we should pay less attention to their international trade.

Additional comments:

Line 56 – “we look at what”

Lines 297-305 – I'm a bit confused by the sampling dates. Earlier it says all of the surveys were conducted in 2023, so how did you “separate responses in four periods”? At another point it says data is between 2010 and 2023. Can you provide clearer information about the temporal information was collected?

Line 157 – “explains why scales were discarded”

Line 191 – “may explain why pangolin researchers and stakeholders believe that international demand for scales is the primary driver of pangolin exploitation in Central and West Africa” I don't think this is the case. Or anyway, it certainly doesn't reflect what I believe.

Reviewer #2 (Remarks to the Author):

This is a thorough, impressive and convincing paper presenting evidence that pangolin exploitation in Nigeria, a key pangolin trafficking hub, is driven by demand for meat for local consumption, rather than by the international scale trade. I enjoyed reading it and believe it makes a highly valuable contribution to our understanding of the drivers of international trade in African pangolins. The paper is highly relevant to broader ongoing debates around illegal wildlife trade and appropriate conservation interventions to counter it, and particularly undermines the prevailing reliance on using international trade bans as a key approach to conserving exploited species.

The organisation, writing, and referencing are extremely high quality, and I have no comments on these. The data is rich and sample sizes are impressive, and the methods and analysis appear sound. The argument for the applicability of results to pangolin exploitation in other C/W African countries is convincing. Figures are clear and informative.

The sole area that I think could be somewhat strengthened, subject to word/reference limits, is the "Implications for pangolin conservation". The below are suggestions for consideration rather than put forward as essential to be considered in revision.

First, it is a serious concern that CITES decisions and rhetoric typically reflect and respond to the perception that international trade (if it occurs) is always driving exploitation, and that CITES listing is going to help the problem. CITES bans are relatively cheap, easy and high profile compared to the difficult, long-term and often obscure efforts to build local alternatives and sound governance that is typically required to address overexploitation driven by other dynamics. Listings/bans are frequently portrayed as "wins", and "job done". This diverts conservation resources and distracts from the real needs for investment. It is a broad problem, not just relevant to pangolins - the dynamic so convincingly set out here is a much more widespread one, whether real drivers of local decline of some CITES-listed species are (e.g.) bushmeat hunting, habitat loss, or human-wildlife conflict. I think more should be made of this argument than the conclusion of the final para, flagging the need for "incorporating domestic drivers of species exploitation into international wildlife trade treaties". I suggest that given the limited scope of action of international wildlife trade treaties, there needs to be more recognition of the limitations of CITES and other trade controls in addressing exploitation driven by local use (or indeed other drivers). This is offered as food for thought rather than a requirement of review.

My second point is that the overall conclusion - that ignoring the actual drivers of exploitation in international trade regulation is problematic - is one that has been argued for decades by analysts in the wildlife trade context, and it would be helpful to locate this conclusion within some key relevant literature. This point was highlighted back in e.g. Hutton and Dickson volume "Endangered Species Threatened Convention" 2000; the Oldfield "Trade in Wildlife" volume 2003. A recent paper (I am an author on this one - but there are many others that could be cited with a bit of digging) explicitly critiques the assumption that "if an internationally traded species faces a level of biological threat, its conservation will benefit from trade restriction" (<https://www.frontiersin.org/journals/ecology-and-evolution/articles/10.3389/fevo.2021.631556/full>).

Minor point- it would be useful to list the IUCN Red List threat status of the focal species.

Reviewer #3 (Remarks to the Author):

The main contribution of the work is the substantial dataset, the collection of which is to be commended. The descriptive analysis that the authors present dispels the common belief that pangolin hunting in parts of Africa is driven by demand for pangolin scales for international trade to Asia. This finding is important for species conservation since it highlights that interventions designed to reduce illegal international wildlife trade will fail to address domestic drivers of pangolin decline.

I have three broad points that I feel need addressing. These relate to suggested interventions, clarity over survey instruments used, and sampling strategies deployed. I provide a brief reflection on each below.

The authors identify three mechanisms through which pangolin hunting & consumption in Nigeria may be reduced (anti-poaching actions, behaviour change programmes for hunters, and efforts to enhance local food security). However, there is no critical engagement with the pros and cons of these different approaches or consideration of the evidence for or against their likely effectiveness. There is a wealth of literature that could be drawn upon for each of the three suggested intervention types.

Whilst I believe the methods used were acceptable (eg a questionnaire was used to gather data from hunters), the methods section needs to more clearly articulate which survey instrument was used to gather data from which groups of people. The authors describe three respondent groups (hunters, vendors & households) and two survey instruments, I think! I believe one survey instrument collected data on what hunters catch & what vendors sell (referred to at L279 as a 'structured interview', and at L297 as a 'structured questionnaire') and a second (i.e., different) survey instrument (mentioned at L343 as 'we interviewed 190 hunters...') was used to collect data on palatability of meat, also from hunters, vendors & households. However, this is not entirely clear. Moreover, it is unclear if a respondent could have been asked to complete both survey instruments (assuming there were indeed two survey instruments), or if the samples are completely independent.

The methods section also lacks clarity on the sampling strategies used to recruit respondents. The authors state that their

analysis quantifying pangolin extraction assumes that their 'community and respondent samples are representative'. Meeting this assumption would require the use of probability-based sampling or census of respondents. It is unclear if this assumption can be met based on the information provided.

Stylistically I think the writing can be strengthened & I have given some comment on a few specific lines as examples.

This following sentence reads as though pangolins have 8 scales: L39 Pangolins, eight scaly African and Asian mammals, are one such taxon threatened by... revision needed.

L55-56 word missing? trade in pangolins scales, we look ?? what motivates local stakeholders...

L52 last sentence of para, language could be strengthened here.

L58 Consider sentence structure. Split into 2 sentences with full stop after ref 19,20 or delete : and replace with were black-bellied (Pha...). Consider the structure of the para. Placing detail of Nigeria's CITES position & levels of pangolin trade before info on your data could strengthen structure here.

L93 unsure what this 'equal weight hunters' means here.

L100 Edit target to past tense.

L126 put in past tense (i.e., throw away).

L157 word missing:explains why scales ?? discarded.

Fig 3A Clarify in legend text if scale & meat prices refer to the 2 species of pangolin combined; presumably red bars show meat prices? Clarify this in legend text.

L175 570 responses or respondents? 570 responses from who? Eg general public, hunters, vendors? Whilst this becomes clear in Fig 4 legend text, it is not clear in main text.

Fig 4 Y axis label could be improved by indicating direction of palatability (e.g., 10 = high palatability).

L203 the text in parentheses describes social desirability bias. It does not address other forms of error as suggested by the inclusion of the word respectively at the end of the sentence. I suggest deleting 'respectively' & the following words on L205 'to these effects'.

L210 consider revising 'suggesting they did not view these as undesirable'. To something along the lines of '...limited sensitivity surrounds this topic'.

L215 sentence wording could be improved here: constitute a by-product of captures intended for meat for local consumption.

L216 wording could be improved/made clearer in sentence starting This is different...

L218 clarify that you are now referring to pangolin meat prices in Nigeria, not Asia.

L224 is the word relatively required?

L237 could 'consumed as meat' be shortened to eaten?

L237 wording could be improved/made clearer in sentence starting First, the proportion that

L253 you recommend anti-poaching actions in protected areas however the evidence of the effectiveness of anti-poaching patrols in reducing illegal resource extraction is mixed.

L253 You identify three mechanisms through which pangolin hunting & consumption in Nigeria may be reduced (anti-poaching actions, behaviour change programmes for hunters, and efforts to enhance local food security). It would be good to see this section expanded. For example, what are the pros and cons of these different approaches; and what evidence is there supporting their adoption in similar contexts; what are the prerequisites for success/effectiveness?

L272 the following text (Hunting is prohibited for certain groups...) reads as though hunting is prohibited for certain groups of people. Perhaps revise to 'Hunting of certain taxa (e.g., x & y) is prohibited...

L277 do you mean pseudo-anonymised? Give that you got written consent, your data are confidential, not anonymous. Please check definitions.

L287 what type of sampling strategy or strategies (they may have differed according to respondent type) were used to recruit hunters & vendors?

L287-L296 please clarify text on sampling strategy, order of content currently confusing & specific form of sampling used (eg snowball sampling; simple random sampling) is not stated. You have three types of respondent (hunter, vendor, households), a clear description of the sampling strategy for each of these types of respondent is needed.

L299 strange sentence order '...here for vendor interviews only the last two sections were used.'

L337 here you state that you assume that your community and respondent samples are representative. Meeting this assumption would require the use of probability-based sampling of respondents. Currently it is unclear if that was the case for your 3 respondent groups (hunters, vendors and households). Indeed, which types of respondents are you referring to with the term 'community and respondent samples?'

It is also confusing at L337 to refer to 'community and respondent samples' given this section is, I believe, describing the analysis of hunter data (ie not the household data).

L343 was the palatability data collected using a different survey instrument to the one described at L279 which you refer to a structured interview (L279)/structured questionnaire (L297)? How were these 3 groups of 190 people recruited? What was the sampling strategy used to identify individuals? Is this 'interview data' collected from different people compared to the structured interview data?

L354-365 Text describing model structure could be clearer & more succinct.

Please provide copies of your questionnaire / interview guides in supporting materials inclusive of your participant information sheets and consent form.

*****END*****

Version 1:

Decision Letter:

30th January 2025

Dear Dr Emogor,

Your manuscript entitled "Pangolin hunting in Southeast Nigeria motivated mainly by local meat consumption not international demand for scales" has now been seen by two of the original three reviewers, whose comments are attached. My apologies for the delay in returning this decision to you. We had been waiting to hear from Reviewer 3, but we have not heard back from them, and we feel able to proceed on the basis of the reviewer input we do have. As you will see, the reviewers are positive about the revisions, but have some remaining concerns, particularly from Reviewer 1, which will need to be addressed before we can offer publication in Nature Ecology & Evolution. We will therefore need to see your responses to the criticisms raised and to some editorial concerns, along with a revised manuscript, before we can reach a final decision regarding publication.

We therefore invite you to revise your manuscript taking into account all reviewer and editor comments. Please highlight all changes in the manuscript text file.

* If you have not done so already please begin to revise your manuscript so that it conforms to our Article format instructions at <http://www.nature.com/natecolevol/info/final-submission>. Refer also to any guidelines provided in this letter.

* Extended Data Figures - please ensure that any supplementary figures and tables that are crucial to the manuscript's conclusions are converted into Extended Data figures and tables to increase visibility of these data. Extended Data figures

and tables are online-only (present in the online PDF and full-text HTML versions of the paper), peer-reviewed display items that provide essential background to the article but are not included in the main article due to space constraints. A maximum of ten Extended Data display items (figures and tables) is permitted.

Link Redacted

Nature Ecology & Evolution is committed to improving transparency in authorship. As part of our efforts in this direction, we are now requesting that all authors identified as 'corresponding author' on published papers create and link their Open Researcher and Contributor Identifier (ORCID) with their account on the Manuscript Tracking System (MTS), prior to acceptance. ORCID helps the scientific community achieve unambiguous attribution of all scholarly contributions. You can create and link your ORCID from the home page of the MTS by clicking on 'Modify my Springer Nature account'. For more information please visit www.springernature.com/orcid.

[redacted]

Reviewer comments:

Reviewer #1 (Remarks to the Author):

I appreciate the revisions from the authors and I feel that the changes go a long way to addressing my concerns. However, I'm still not fully convinced by the main argument presented, specifically the "assertion that pangolin stakeholders perceive international demand for scales as the primary driver of African pangolin exploitation". The evidence for this is in the "scoping study" cited. In the scoping study, on page 78 as cited in the rebuttal, stakeholders were asked about "threats" and scored "international demand" (5) as higher than household consumption or subsistence commercialization, scored generally as (3). Again, I don't find this as contradictory at all to the findings from this study that the primary motivation for hunting is meat. Both can be true! The primary motivation for hunters can be meat and the threat to pangolin persistence could still be demand for scales more than meat.

Specifically, the line that "Despite this, a subset of pangolin researchers and stakeholders in Central and West Africa (n = 55 people) believe that international demand for pangolin scales is a stronger driver of pangolin hunting in these regions than local use". This is not what the scoping study suggests. Stakeholders were asked about threats. And yes, the stakeholders feel that international demand is a greater threat than meat. The data presented here doesn't disprove this. The data only show that hunters are motivated more by meat than scales. This is important, and it helps to contextualize the dynamics surrounding pangolin exploitation. I don't deny the importance of these results. But I don't think this should be taken as evidence that scales are less of a threat to pangolins than meat.

It all comes down to change over time, exploitation vs. overexploitation, and how motivation scales up (or down) into the larger economic picture. First of all, are Cross River pangolins overexploited at all? This isn't really made clear in the paper. It's implied that they are in paragraph one but this is not made explicit. If the pangolins are not overexploited, i.e. they're simply being exploited, then I suppose it doesn't matter too much from a conservation perspective what exactly motivates the hunters. As stated in the paper, "African pangolins have been exploited long before being trafficked to Asia, with their exploitation tied to rural communities' use of wildlife to supplement food and income". This is so critical, in my view, for contextualizing the conservation problem here. These populations have been subject to strong hunting pressure for a long time. So, are these populations now overexploited and/or in danger of extinction? My guess is that they are... and if so, there has to be a reason underlying this. The added incentive of scales (even if secondary to meat consumption) is one potential culprit. Increased human population sizes and or demand for meat is another. We don't know. But if pangolins are threatened here then something must have changed, because presumably their palatability has not. Without knowing this, I think we need to be really careful about changing broad/global pangolin conservation strategies on the basis of these findings – otherwise we risk making the same mistakes as the authors suggest other pangolin conservationists are making at the moment.

By the way, let's look at the campaigns from conservation organizations as noted in the rebuttal (although, I generally feel that this is weak evidence for characterizing scientific consensus on issues). The WWF website states that "They certainly are one of the most trafficked mammals in Asia and, increasingly, Africa. Pangolins are in high demand in countries like

China and Vietnam. Their meat is considered a delicacy and pangolin scales are used in traditional medicine and folk remedies to treat a range of ailments from asthma to rheumatism and arthritis.” Based on the results here, how should this be changed to better reflect the importance of meat driving the hunting of pangolins? I don’t necessarily see this as a problematic “scales-centred narrative”. The TRAFFIC website states that “As with many other wildlife species, the motivations behind consumption of pangolin products varies significantly between nations and social groups.” This seems pretty consistent with the results from this study. Yes, there are some overly simplistic statements about a “global trade to feed into” as noted on the Flora & Fauna site. But I don’t think this constitutes evidence for scientific consensus – there are a range of reasons for writing a statement like this on a donation page that are not science-related.

Anyway, to be clear, I’m arguing all of these points not because I don’t think this work should be published or that I don’t think this is important. I just hope that the findings are used in an appropriate way that effectively supports the conservation of pangolins, in Nigeria and more broadly in Africa and Asia.

I have a few suggested edits along these lines:

As raised in the rebuttal, “Additionally, our research centres on the ‘relative importance or contribution’ of scales and meat in driving pangolin exploitation (in other words, which is more important) – not whether one product is important and the other not.”

If this is the case, then I would suggest changing the title to:

“Pangolin hunting in Southeast Nigeria motivated more by local meat consumption than international demand for scales”

The present title definitely gives me the impression that pangolin hunting in Southeast Nigeria is “not” motivated by international demand for scales.

I would also suggest changing this Introduction sentence to:

“...there is a widespread assertion among pangolin researchers and stakeholders in Central and West Africa that international demand for scales is the primary threat faced by pangolins in these regions”

This may require then better connecting the issues of “threats” and hunting “motivation”.

Otherwise, the other revisions in response to my original review are good and go a long way to clarifying some of my initial concerns. The Discussion in particular is much stronger now in my opinion. I think it’s pretty clear that we all agree that knowing the relative drivers of pangolin exploitation are important for management and action planning. The Discussion now well covers how these results support this.

Additional comments:

Lines 63-66 – “In a mixed scenario, where hunters primarily target pangolins for meat but trade scales as a by-product (or vice versa), the most effective interventions are likely to be those that address the primary product behind their exploitation.” I disagree. I suspect the best interventions in this case would be those that address both motivations underlying the exploitation.

Line 210 – “may explain why”

Line 211 – again, as noted above, be careful with wording here

Lines 246-248 – as highlighted in the Introduction, pangolins here have been hunted for a long time – so what’s changed or HAS anything changed? Is their “survival” threatened here at all?

Reviewer #1 (Remarks on code availability):

I briefly reviewed the code and data. Everything appears to be there and appropriately annotated.

Reviewer #2 (Remarks to the Author):

Many thanks for addressing my comments. I am happy with the response and suggest just two very minor points:

line 297: "Looking beyond pangolins, our results underscore the importance of incorporating CONSIDERATION OF domestic drivers of species exploitation into DECISIONMAKING WITHIN international wildlife trade treaties". I suggest adding the capitalised text because by their very nature, international wildlife trade treaties can't generally address domestic drivers of exploitation - that generally falls within national jurisdiction. However, decisionmaking within these regimes can explicitly consider domestic drivers of exploitation.

line 306: "drives" should be "drivers".

*****END*****

Version 2:

Decision Letter:

11th February 2025

Dear Dr. Emogor,

Thank you for submitting your revised manuscript "Pangolin hunting in Southeast Nigeria motivated more by local meat consumption than international demand for scales" (NATECOLEVOL-24092457B). It has now been seen again by the original reviewers and their comments are below. The reviewer has no remaining comments, and therefore we'll be happy in principle to publish it in Nature Ecology & Evolution, pending minor revisions to comply with our editorial and formatting guidelines.

[redacted]

Reviewer #1 (Remarks to the Author):

The authors have satisfactorily addressed my concerns, thanks. I'm sure this will be a widely read and key paper for understanding pangolin exploitation.

Response to Reviewers (NATECOLEVOL-24092457)

Reviewer #1

The authors present important data from hunters and vendors in Nigeria on the trade of pangolins. Indeed, as a major hub in the international trafficking of pangolins, such data is important for context in the conservation of these important species. In general, I feel that the manuscript is well written, the methods are robust, and the results quite clear.

Thank you very much for reviewing our work and for your positive overall assessment. Your comments have helped us increase the manuscript's clarity and rigour. We have responded to the comments below – all referenced line numbers refer to those in the clean version of the manuscript (new texts are highlighted).

However, I feel that there is a strawman argument presented that I don't find convincing. The suggestion that 'pangolin researchers believe that international demand for scales is a primary driver of pangolin exploitation' just doesn't ring true to me. I'm a pangolin researcher and this has never been my belief. It simplifies what I believe to be a fairly complex issue. As stated clearly in the paper, African pangolins have been hunted for local community use for, presumably, as long as those communities have been there. We know from the data that the pangolins are and have been hunted for their meat. If, as suggested by the paper, the scales are not important in driving hunting here then that could mean a few things as I see it: 1) these pangolin populations haven't changed much and they're not in fact overexploited... they're hunted for their meat as they have been for centuries, 2) even though scales are not important, hunting has increased recently, perhaps because human population sizes have increased. Otherwise, I think it's important to consider another option consistent with the results here: 3) the scales are in fact important – while hunting for scales is not the "primary" reason to hunt pangolins it still may provide sufficient incentive to increase hunting rates into a state of overexploitation.

My issue with the paper is that the data doesn't provide evidence for or against either of these possibilities, but the primary argument of the paper is against option 3. But I don't think that showing 1) the price of scales is lower than meat, nor 2) most hunters hunt for the meat, necessarily negate this possibility. From Fig 1, we know that the price of scales is ~1500 NGN while meat is ~5000 NGN. That's still presumably decent money? This is evident from the fact that 1/3 of the scales are sold. In the paper this is presented as "only about one-third of scales were sold". If 20,000 pangolins are hunted, and the scales from 6,000 of those enter the market, this strikes me as potentially concerning. It could be sufficiently added incentive, on top of the primary motivation to hunt them for meat, to lead to overexploitation. I don't know if this is the case or not... but to use the data here to suggest that this is NOT the case appears flawed to me.

Simo et al. Oryx 2023 has similar data which I find instructive for comparison. Prices for scales, for white and black bellied, reported in that study are similar to this one but the scales of giant pangolin were more than “15 times the value estimated for the scales of white-bellied pangolin”. So, it’s worth noting that there are likely some species-specific differences here with respect to meat vs scale prices. Also, while the price results in that study and this one are similar, Simo et al. found evidence indeed of bespoke supply chains for scales. My best guess behind this difference is the fact there is likely to be strong regional variation in these sorts of things (but this undermines the “Looking beyond Nigeria” result to some extent).

Of course, I agree that we need more attention on what’s happening “on the ground” with hunters/vendors and, as noted earlier, I do find great value in the data collected. But I don’t think there is good evidence presented to support the idea that scales are not important and that we should pay less attention to their international trade.

Thank you for your thorough assessment of our findings and conclusions. We have addressed your three main points below: 1) our assertion that pangolin stakeholders perceive international demand for scales as the primary driver of African pangolin exploitation, 2) the lack of clarity regarding the scenario reflected by our results, and 3) the possibility that our findings may be species-specific.

On the first point, the text “...there is a widespread assertion among pangolin researchers and stakeholders in Central and West Africa that international demand for scales is the primary driver of pangolin hunting in these regions” is based on a report that presented the viewpoints of 55 stakeholders (including researchers) in Central and Western Africa on threats to pangolins. See pages 6 and 78 of the report (accessible via https://pdf.usaid.gov/pdf_docs/PA00X9WW.pdf). The framing and focus of the following global campaigns by renowned conservation organisations also point to the prevalence of the scales-centred narrative 1) <https://www.worldwildlife.org/species/pangolin>, 2) <https://www.traffic.org/what-we-do/species/pangolins/>, 3) <https://www.fauna-flora.org/appeals/save-pangolins/> (“Without a global trade to feed into, there will be almost no reason for poachers to take pangolins from the wild”), 3) <https://secure.wcs.org/donate/help-save-pangolins>

Nonetheless, we acknowledge that our wording may suggest that all pangolin stakeholders share this view and have therefore clarified our perception that the belief is held by a subset of stakeholders. We have also referenced other research that looked at the exploitation of pangolins at site levels while highlighting the gap our work aims to fill:

“Although existing research has characterised local dynamics of African pangolin exploitation^{15–17,19,20}, there is still no explicit assessment of the proximate drivers of African pangolin exploitation – i.e., how far hunters are motivated primarily by the international demand for scales, local demand for meat, or a combination of both markets and products. Despite this, a subset of pangolin researchers and stakeholders in Central and West Africa (n = 55 people) believe that international demand for pangolin scales is a stronger driver of pangolin hunting in these regions than local use²¹” – lines 49-55

On the second point, we want to start by noting that what we suggest is complementary priority on more localised actions and not less focus on international trade:

“CITES and other stakeholders should support pangolin range countries to assess local drivers of exploitation, and where local exploitation drivers differ from international trade, bespoke interventions tackling domestic exploitation should be designed and implemented.” (in lines 305-308).

We have stressed the same point again:

“Therefore, and consistent with our findings, focusing on trade restrictions without complementary local measures to curb exploitation in and around species habitats may prove ineffective.” (lines 301-303).

Additionally, our research centres on the ‘relative importance or contribution’ of scales and meat in driving pangolin exploitation (in other words, which is more important) – not whether one product is important and the other not. Our framing of ‘relative importance’ aligns with our research focus on how much the demand for the different products (i.e., meat and scales) drives local pangolin exploitation. Therefore, by showing that scales for international demand are not the primary driver of pangolin exploitation, we are not discounting the high mass of scales traded (involving roughly 7,000 individuals from the Cross River region annually, based on our estimates) – but we are saying that the commodity is not as important as meat in driving the exploitation of pangolins in Cross River.

The scenarios you have provided are indeed all plausible, but our results point more closely to a fourth scenario: that the demand for scales is not the primary driver of pangolin exploitation in Southeast Nigeria, but scales from pangolins hunted for meat are still sold (as a by-product of meat-centred hunting). We believe we have provided good evidence in support of this scenario: 1) demand for meat is the primary motivation for killing pangolins, 2) targeted pangolin hunting is rare, which is not what one would expect if the animals were killed for scales, 3) meat fetches a substantially higher price locally than scales. As stated in the manuscript, while our data are prone to desirability bias, we show that they align with quantitative data independently gathered from the same landscape. Furthermore, as Reviewers 2 and 3 point out, recognising that meat and not scales is the primary driver is crucial, given the profound consequences of focusing on international trade restrictions without implementing complementary actions around pangolin habitats.

Notwithstanding, we agree that scales may serve as an added incentive to pangolin hunting in future and have added the following text to the Discussion:

“We also appreciate that, although scales may currently be a by-product of pangolin hunting in southeast Nigeria, in the long-term – especially without appropriate conservation actions – a more established market for the commodity could emerge, which may incentivise hunters to target pangolins specifically to meet international demand for scales.” – lines 262-266

We also added a new paragraph in the Introduction to clarify our point about the conservation implications of the different scenarios:

“Here, against the backdrop of intense focus on international trade in pangolin scales, we look at what motivates local stakeholders to kill and trade pangolins. Because these may have divergent implications for conservation actions, it is essential to identify the relative importance of scales and meat in driving pangolin exploitation. If pangolin hunting is driven by international demand for scales, conservation efforts could focus on disrupting trade networks, enforcing CITES regulations, and running demand-reduction campaigns in consumer countries. Conversely, if local demand for meat is the main driver, prioritizing community engagement through alternative livelihoods or education campaigns may be more effective. In a mixed scenario, where hunters primarily target pangolins for meat but trade scales as a by-product (or vice versa), the most effective interventions are likely to be those that address the primary motivations behind their exploitation.” – lines 56-66

On the last point, we agree that our findings are relevant to black- and white-bellied pangolins only and have added a caveat in the Discussion:

“Further, our two focal species, averaging around 2 kg in weight, are considerably smaller than other African pangolin species^{27,28} (approximately 31 kg for giant pangolins and 10 kg for Temminck’s pangolins; *Smutsia temminckii*^{29,30}). As a result, our findings may be specific to black- and white-bellied pangolins (*Phataginus* spp.). However, this does not diminish the importance of our results as *Phataginus* spp. represent approximately 98% of African pangolins trafficked internationally (based on seizure data)⁵ and 96% of pangolins caught by hunters in Central and West Africa (based on hunter offtake data from six countries)¹⁵.” – lines 230-236

We have also revised our title to indicate that our results primarily apply to southeast Nigeria:

“Pangolin hunting in Southeast Nigeria motivated mainly by local meat consumption not international demand for scales”

Additional comments:

Line 56 – “we look at what”

Thank you for spotting the typo. We have added “at”. – line 57

Lines 297-305 – I’m a bit confused by the sampling dates. Earlier it says all of the surveys were conducted in 2023, so how did you “separate responses in four periods”? At another point it says data is between 2010 and 2023. Can you provide clearer information about the temporal information was collected?

We did not collect data at other times but asked many of our questions for different periods, beginning in 2010. We have clarified this in the Methods – lines 355-360

Line 157 – “explains why scales were discarded”

We have added “were” – line 173

Line 191 – “may explain why pangolin researchers and stakeholders believe that international demand for scales is the primary driver of pangolin exploitation in Central and West Africa” I don’t think this is the case. Or anyway, it certainly doesn’t reflect what I believe.

Here, we attempt to make sense of the WABiCC/USAID report cited above using a suggestive tone (i.e., ‘may’). Nonetheless, as in the Introduction, we have clarified that this belief is not widespread among all pangolin researchers and stakeholders:

“The higher mass of scales compared to meat in the illegal pangolin trade ^{5,7} may explain why a many pangolin researchers and stakeholders believe that international demand for scales is a stronger driver of pangolin hunting in Central and West Africa than local use ²¹” – **lines 209-211**

Thank you again for your time and helpful comments!

Reviewer #2

This is a thorough, impressive and convincing paper presenting evidence that pangolin exploitation in Nigeria, a key pangolin trafficking hub, is driven by demand for meat for local consumption, rather than by the international scale trade. I enjoyed reading it and believe it makes a highly valuable contribution to our understanding of the drivers of international trade in African pangolins. The paper is highly relevant to broader ongoing debates around illegal wildlife trade and appropriate conservation interventions to counter it, and particularly undermines the prevailing reliance on using international trade bans as a key approach to conserving exploited species. The organisation, writing, and referencing are extremely high quality, and I have no comments on these. The data is rich, and sample sizes are impressive, and the methods and analysis appear sound. The argument for the applicability of results to pangolin exploitation in other C/W African countries is convincing. Figures are clear and informative.

Thank you very much for reviewing our work and providing constructive comments to help improve it. We are delighted with your positive feedback and have addressed your comments below – all referenced line numbers refer to those in the clean version of the manuscript (new texts are highlighted).

The sole area that I think could be somewhat strengthened, subject to word/reference limits, is the “Implications for pangolin conservation”. The below are suggestions for consideration rather than put forward as essential to be considered in revision. First, it is a serious concern that CITES decisions and rhetoric typically reflect and respond to the perception that international trade (if it occurs) is always driving exploitation, and that CITES listing is going to help the problem. CITES bans are relatively cheap, easy and high profile compared to the difficult, long-term and often obscure efforts to build local alternatives and sound governance that is typically required to address overexploitation driven by other dynamics. Listings/bans are frequently portrayed as “wins”, and “job done”. This diverts conservation resources and distracts from the real needs for investment. It is a broad problem, not just relevant to pangolins - the dynamic so convincingly set out here is a much more widespread one, whether real drivers of local decline of some CITES-listed species are (e.g.) bushmeat hunting, habitat loss, or human-wildlife conflict. I think more should be made of this argument than the conclusion of the final para, flagging the need for “incorporating domestic drivers of species exploitation into international wildlife trade treaties”. I suggest that given the limited scope of action of international wildlife trade treaties, there needs to be more recognition of the limitations of CITES and other trade controls in addressing exploitation driven by local use (or indeed other drivers). This is offered as food for thought rather than a requirement of review.

My second point is that the overall conclusion - that ignoring the actual drivers of exploitation in international trade regulation is problematic - is one that has been argued for decades by analysts in the wildlife trade context, and it would be helpful to locate this conclusion within some key relevant literature. This point was highlighted back in e.g. Hutton and Dickson volume “Endangered Species Threatened Convention” 2000; the Oldfield “Trade

in Wildlife” volume 2003. A recent paper (I am an author on this one - but there are many others that could be cited with a bit of digging) explicitly critiques the assumption that “if an internationally traded species faces a level of biological threat, its conservation will benefit from trade restriction” (<https://www.frontiersin.org/journals/ecology-and-evolution/articles/10.3389/fevo.2021.631556/full>).

We have addressed both these points in the last paragraph of the Discussion, drawing on your paper recommendations:

“Looking beyond pangolins, our results underscore the importance of incorporating domestic drivers of species exploitation into international wildlife trade treaties³⁶. Among taxa with biological resource use as a threat, four times more species are threatened by local use than international trade³⁷. Our results thus highlight an important pathway where international trade regulations may not reduce exploitation pressures as supplying international trade may not be the primary reason species are hunted. Therefore, and consistent with our findings, focusing on international trade restrictions without complementary local measures to curb exploitation in and around species habitats may prove ineffective^{40,42}. This consideration is especially crucial for CITES Appendix II species, whose commercial trade is permitted if the trade does not harm the species’ survival in the wild. CITES and other stakeholders should support pangolin range countries to assess local drives of exploitation, and where local exploitation drivers differ from international trade, bespoke interventions tackling domestic exploitation should be designed and implemented.” – lines 296-308

Minor point- it would be useful to list the IUCN Red List threat status of the focal species.

We have provided the IUCN categories of the different pangolin species:

“Black-bellied pangolins are classified as Vulnerable, while white-bellied and giant pangolins are listed as Endangered on the IUCN Red List (International Union for Conservation of Nature)⁴³” – lines 76-77.

Thank you again for your very helpful comments!

Reviewer #3

The main contribution of the work is the substantial dataset, the collection of which is to be commended. The descriptive analysis that the authors present dispels the common belief that pangolin hunting in parts of Africa is driven by demand for pangolin scales for international trade to Asia. This finding is important for species conservation since it highlights that interventions designed to reduce illegal international wildlife trade will fail to address domestic drivers of pangolin decline.

Thank you very much for reviewing our work and for your positive feedback. We have addressed all your comments, which has helped to increase the clarity and rigour of our manuscript. Please note that all referenced line numbers refer to those in the clean version of the manuscript (new texts are highlighted).

I have three broad points that I feel need addressing. These relate to suggested interventions, clarity over survey instruments used, and sampling strategies deployed. I provide a brief reflection on each below.

The authors identify three mechanisms through which pangolin hunting & consumption in Nigeria may be reduced (anti-poaching actions, behaviour change programmes for hunters, and efforts to enhance local food security). However, there is no critical engagement with the pros and cons of these different approaches or consideration of the evidence for or against their likely effectiveness. There is a wealth of literature that could be drawn upon for each of the three suggested intervention types.

We agree that further assessing the merits and caveats of the proposed actions is crucial and have now added the following to the Discussion:

“Instead, priority should be given to site-level interventions, such as anti-poaching patrols and community based actions, including initiatives to improve food security and behaviour-change programmes for hunters. Patrols can deter poaching and reduce threats to pangolins via snare removals³⁹. However, they are mostly confined to protected areas and are often limited in their effectiveness⁴⁰ because of insufficient resources to ensure adequate patrol effort⁴¹. Behaviour-change interventions combined with food security programs are likely to be effective and well-received by local communities, as they address direct needs without antagonising them^{18,42}. However, caution is needed during project design to ensure compliance and that hunting does not shift to other threatened species⁴³.” – lines 286-295

Whilst I believe the methods used were acceptable (eg a questionnaire was used to gather data from hunters), the methods section needs to more clearly articulate which survey instrument was used to gather data from which groups of people. The authors describe three respondent groups (hunters, vendors & households) and two survey instruments, I think! I

believe one survey instrument collected data on what hunters catch & what vendors sell (referred to at L279 as a ‘structured interview’, and at L297 as a ‘structured questionnaire’) and a second (i.e., different) survey instrument (mentioned at L343 as ‘we interviewed 190 hunters...’) was used to collect data on palatability of meat, also from hunters, vendors & households. However, this is not entirely clear. Moreover, it is unclear if a respondent could have been asked to complete both survey instruments (assuming there were indeed two survey instruments), or if the samples are completely independent.

Many thanks for this point. We agree that the data collection protocol could be streamlined and have revised that section (please see ‘Data Collection’). We began the section by noting the data types as follows:

“Data Collection

This section outlines the data collection protocols for our two surveys. The main survey focused on hunters and vendors (809 respondents across 32 locations; hunter and vendor behaviour survey), while the second survey involved hunters, vendors, and household members (570 respondents across 15 locations; palatability survey). We conducted the surveys a year apart, but the datasets are not entirely independent, as 11 locations targeted in the hunter and vendor behaviour survey were also used for the palatability survey (33% overlap).” – lines 324-330

The methods section also lacks clarity on the sampling strategies used to recruit respondents. The authors state that their analysis quantifying pangolin extraction assumes that their ‘community and respondent samples are representative’. Meeting this assumption would require the use of probability-based sampling or census of respondents. It is unclear if this assumption can be met based on the information provided.

We have clarified, in the Survey Data section, the protocol we used to gather the data. More details about the protocol for the palatability data are in the referenced preprint. Regarding representativeness, we assume (when extrapolating to annual landscape-level pangolin offtake) that our samples are representative. The section now reads:

“Next, we multiplied these values by the median number of hunters in each community (per hunter category), which we derived through our household census (see Data Collection). We then multiplied these estimated totals per community by the total number of communities in the landscape; Table S3) – assuming that the number of hunters in our focal communities (obtained via the census) is representative of that of other communities in the landscapes” – lines 403-407

We thought it was reasonable to assume representativeness for the following reasons: 1) we selected most of our communities using stratified random sampling, and 2) we took a comprehensive approach to gathering data from a sample all relevant households – by first identifying all households in our sample communities whose members were hunters and vendors and then interviewing a large proportion of them.

Stylistically I think the writing can be strengthened & I have given some comment on a few specific lines as examples.

This following sentence reads as though pangolins have 8 scales: L39 Pangolins, eight scaly African and Asian mammals, are one such taxon threatened by... revision needed.

Revised as advised:

“Pangolins, eight species of scaly African and Asian mammals, are one such taxon threatened by overexploitation across their range as well as internationally⁴.” – lines 39-40

L55-56 word missing? trade in pangolins scales, we look ?? what motivates local stakeholders...

Thank you for spotting the typo. We have added “at”. – line 57

L52 last sentence of para, language could be strengthened here.

We have revised the sentence as follows:

“Although existing research has characterised local dynamics of African pangolin exploitation^{15–17,19,20}, there is still no explicit assessment of the proximate drivers of African pangolin exploitation – i.e., whether hunters are motivated primarily by the international demand for scales, local demand for meat, or a combination of both markets and products.” – line 49-52

L58 Consider sentence structure. Split into 2 sentences with full stop after ref 19,20 or delete : and replace with were black-bellied (Pha...). Consider the structure of the para. Placing detail of Nigeria’s CITES position & levels of pangolin trade before info on your data could strengthen structure here.

Revised as advised:

“To test the prevalence of these scenarios, we gather data from hunters, wild meat market vendors, and household members in Nigeria’s Cross River Forest landscape. Nigeria is a signatory to CITES and the biggest trade hub for pangolin trafficking globally, with seizures over a 11-year period (2010—2021) involving over 190,000 kg of scales from an estimated ~800,000 African pangolins⁵. Hunting, trading, and consuming pangolins are illegal in Nigeria^{18,23}, but these practices are common, including around Cross River landscape – a pangolin stronghold and poaching hotspot^{21,22}. Black-bellied (*Phataginus tetradactyla*) and white-bellied pangolins (*P. tricuspis*) occur there, with giant pangolins (*Smutsia gigantea*) occurring in adjoining Cameroonian forests. Black-bellied pangolins are classified as Vulnerable, while white-bellied and giant pangolins are listed as Endangered on the IUCN Red List (International Union for Conservation of Nature)⁴.” – lines 67-77

L93 unsure what this ‘equal weight hunters’ means here.

That was a typo, which has been addressed by deleting ‘hunters’:

“When asked how pangolins were captured, formal hunters told us that simply picking them up by hand was the most common method, with a mean across hunters of 89% (median of 100%) of all formal hunters’ captures (compared with 6% by trap, 3% using a dog and 2% using a gun; all responses were given equal weight here and in other calculations).” – lines 105-108

L100 Edit target to past tense.

Revised as advised – lines 135

L126 put in past tense (i.e., throw away).

Revised as advised – lines 142

L157 word missing: ...explains why scales ?? discarded.

We have corrected the error:

“The low price of scales, coupled with an apparent ineffective or nascent supply chain, likely explains why scales were discarded.” – lines 172-173

Fig 3A: Clarify in legend text if scale & meat prices refer to the 2 species of pangolin combined; presumably red bars show meat prices? Clarify this in legend text.

We have added the following texts:

“Fig. 3: Trends in the price paid to hunters and vendors for pangolin scales and meat, and the meat of three other commonly hunted species. (a) The real price of all the scales from individual adult black- and white-bellied pangolins was lower than that of their meat, with meat prices falling over time more steeply than scale prices – the panel shows combined prices for both pangolin species. (b-d) The trend in pangolin meat price was comparable with that for three other commonly harvested species – African brush-tailed porcupine, blue duiker, and red river hog. The error bars show the effects of the respective variables – the circles are model predictions, while the vertical lines are 95% confidence intervals. The greenish-blue bars represent meat prices, while the brown bars indicate scale prices” – lines 180-187

L175: 570 responses or respondents? 570 responses from who? Eg general public, hunters, vendors? Whilst this becomes clear in Fig 4 legend text, it is not clear in main text.

We have clarified this in the text, including in the ‘Survey Data’ section:

“Our independent palatability dataset gathered from 570 hunters, vendors, and other household members from the Cross River forest landscape showed that pangolins have the highest palatability compared to all wild and domestic meat, fish, or invertebrates that we asked about (Fig. 4).” – lines 193-196

Fig 4: Y axis label could be improved by indicating direction of palatability (e.g., 10 = high palatability).

We have revised the plot – line 201

L203: The text in parentheses describes social desirability bias. It does not address other forms of error as suggested by the inclusion of the word respectively at the end of the sentence. I suggest deleting ‘respectively’ & the following words on L205 ‘to these effects’.

Deleted as advised.

L210: Consider revising ‘suggesting they did not view these as undesirable’. To something along the lines of ‘...limited sensitivity surrounds this topic’.

Revised as advised:

“3) our respondents freely admitted conducting illegal activities, including killing and trading pangolins, suggesting minimal sensitivity around this topic.” – lines 228-230

L215: Sentence wording could be improved here: constitute a by-product of captures intended for meat for local consumption.

Revised as advised:

“Rather, it seems that those scales which are trafficked from the landscape are a by-product of pangolins captured for their meat for local consumption.” – lines 239-241

L216: wording could be improved/made clearer in sentence starting “This is different...

Revised as advised:

“Note that the consumption of pangolin meat across Sub-Saharan Africa, and certainly in our study location different from many rural Asian communities who are increasingly selling pangolin meat to cities where eating meat commands signals high social status³⁰ and is therefore a luxurious activity^{8,11}.” – lines 241-244

L218: clarify that you are now referring to pangolin meat prices in Nigeria, not Asia.

Thank you. We have clarified as advised:

“Further, the higher price of pangolin meat than scales in our study indicates that meat may be in greater demand, with this demand likely driven by pangolin meat’s exceptional palatability.” – lines 244

L224: is the word relatively required?

‘Relatively’ deleted.

L237: could ‘consumed as meat’ be shortened to eaten?

Revised as advised.

L237: wording could be improved/made clearer in sentence starting “First, the proportion that

Revised as advised:

“First, the proportion of pangolins in the overall hunter offtake in our study landscape (approximately 2%)²⁵ is similar to figures reported across these regions – based on data from sites in Cameroon, Central African Republic, Democratic Republic of Congo, Equatorial Guinea, Gabon, and the Republic of Congo¹⁵.” – lines 270-273

L253: you recommend anti-poaching actions in protected areas however the evidence of the effectiveness of anti-poaching patrols in reducing illegal resource extraction is mixed.

We agree that the evidence is mixed and have highlighted that in the Discussion. We believe, however, that there is a need to protect pangolins in their natural habitats and patrols may be effective in doing so, especially when combined with community-based actions.

L253: You identify three mechanisms through which pangolin hunting & consumption in Nigeria may be reduced (anti-poaching actions, behaviour change programmes for hunters, and efforts to enhance local food security). It would be good to see this section expanded. For example, what are the pros and cons of these different approaches; and what evidence is there supporting their adoption in similar contexts; what are the prerequisites for success/effectiveness?

Thanks for this very important point. We have now added the following texts:

“Instead, priority should be given to site-level interventions, such as anti-poaching patrols and community based actions, including initiatives to improve food security and behaviour-change programmes for hunters. Patrols can deter poaching and reduce threats to pangolins via snare removals³⁹. However, they are mostly confined to protected areas and are often limited in their effectiveness⁴⁰ because of insufficient resources to ensure adequate patrol effort⁴¹. Behaviour-change interventions combined with food security programs are likely to be effective and well-received by local communities, as they address direct needs without antagonising them^{18,42}. However, caution is needed during project design to ensure compliance and that hunting does not shift to other threatened species.” – lines 288-295

In line with this point, we also added a new paragraph in the Introduction to give more context to the conservation implications of the possible scenarios driving pangolin exploitation:

“Here, against the backdrop of intense focus on international trade in pangolin scales, we look at what motivates local stakeholders to kill and trade pangolins. Because these may have divergent implications for conservation actions, it is essential to identify the relative importance of scales and meat in driving pangolin exploitation. If pangolin hunting is driven by international demand for scales, conservation efforts could focus on disrupting trade networks, enforcing CITES regulations, and running demand-reduction campaigns in consumer countries. Conversely, if local demand for meat is the main driver, prioritizing community engagement through alternative livelihoods or education campaigns may be more effective. In a mixed scenario, where hunters primarily target pangolins for meat but trade scales as a by-product (or

vice versa), the most effective interventions are likely to be those that address the primary motivations behind their exploitation.” – lines 56-66

L272: The following text (Hunting is prohibited for certain groups...) reads as though hunting is prohibited for certain groups of people. Perhaps revise to ‘Hunting of certain taxa (e.g., x & y) is prohibited...

Revised as advised.

L277: do you mean pseudo-anonymised? Give that you got written consent, your data are confidential, not anonymous. Please check definitions.

All the data are completely anonymised. Written consents did not include the names of participants – they only signed the forms. While a signature is still personal information, it was not part of the data collected for the study, and it is impossible to link the consent forms to each participant.

L287: What type of sampling strategy or strategies (they may have differed according to respondent type) were used to recruit hunters & vendors?

Thanks for this. We have clarified recruitment methods for the different groups in the “Data Collection” section. In summary, we recruited hunters and vendors in their homes – we visited all the houses in each community to identify households whose members were hunters and vendors and later returned to those households to conduct interviews. We then supplemented the data from vendors via visits to wild meat markets in the community.

L287-L296: Please clarify text on sampling strategy, order of content currently confusing & specific form of sampling used (eg snowball sampling; simple random sampling) is not stated. You have three types of respondents (hunter, vendor, households), a clear description of the sampling strategy for each of these types of respondent is needed.

Thank you. Similar to the point above, we have provided additional detail in the “Data Collection” section.

L299: Strange sentence order ‘...here for vendor interviews only the last two sections were used.’

We have revised the phrase:

“the questionnaire for vendors contained only sections e and f.” – lines 354-355

L337: Here you state that you assume that your community and respondent samples are representative. Meeting this assumption would require the use of probability-based sampling of respondents. Currently it is unclear if that was the case for your 3 respondent groups (hunters, vendors and households). Indeed, which types of respondents are you referring to with the term ‘community and respondent samples?’ It is also confusing at L337 to refer to ‘community and respondent samples’ given this section is, I believe, describing the analysis of hunter data (ie not the household data).

We addressed this point above. By ‘community and respondent samples’ we mean the number of hunters we counted in our focal communities:

“Next, we multiplied these values by the median number of hunters in each community (per hunter category), which we derived through our household census (see Data Collection). We then multiplied these estimated totals per community by the total number of communities in the landscape; Table S3) – assuming that the number of hunters in our focal communities (obtained via the census) is representative of other communities in the landscapes.” – lines 403-407

L343 was the palatability data collected using a different survey instrument to the one described at L279 which you refer to a structured interview (L279)/structured questionnaire (L297)? How were these 3 groups of 190 people recruited? What was the sampling strategy used to identify individuals? Is this ‘interview data’ collected from different people compared to the structured interview data?

Palatability data were gathered via a different survey – see ‘Palatability Survey’ subsection where we summarise our data collection protocol. Details on how the data were gathered are in the referenced manuscript. We could not provide additional information due to word restrictions.

L354-365 Text describing model structure could be clearer & more succinct.

We have attempted to edit the modelling section for clarity.

Please provide copies of your questionnaire / interview guides in supporting materials inclusive of your participant information sheets and consent form.

We have provided these items in the Supplementary Material.

Thank you again for your very helpful comments!

Response to Reviewers (NATECOLEVOL-24092457-R2)

Reviewer #1

I appreciate the revisions from the authors, and I feel that the changes go a long way to addressing my concerns.

Thank you for reviewing our work again. We appreciate your time, thoughtful feedback, and positive assessment of the previous version. We have incorporated all your suggestions and revised the manuscript accordingly, with new text highlighted (line numbers refer to the track changes version).

However, I'm still not fully convinced by the main argument presented, specifically the "assertion that pangolin stakeholders perceive international demand for scales as the primary driver of African pangolin exploitation". The evidence for this is in the "scoping study" cited. In the scoping study, on page 78 as cited in the rebuttal, stakeholders were asked about "threats" and scored "international demand" (5) as higher than household consumption or subsistence commercialization, scored generally as (3). Again, I don't find this as contradictory at all to the findings from this study that the primary motivation for hunting is meat. Both can be true! The primary motivation for hunters can be meat and the threat to pangolin persistence could still be demand for scales more than meat.

Specifically, the line that "Despite this, a subset of pangolin researchers and stakeholders in Central and West Africa (n = 55 people) believe that international demand for pangolin scales is a stronger driver of pangolin hunting in these regions than local use". This is not what the scoping study suggests. Stakeholders were asked about threats. And yes, the stakeholders feel that international demand is a greater threat than meat. The data presented here doesn't disprove this. The data only show that hunters are motivated more by meat than scales. This is important, and it helps to contextualize the dynamics surrounding pangolin exploitation. I don't deny the importance of these results. But I don't think this should be taken as evidence that scales are less of a threat to pangolins than meat.

It all comes down to change over time, exploitation vs. overexploitation, and how motivation scales up (or down) into the larger economic picture. First of all, are Cross River pangolins overexploited at all? This isn't really made clear in the paper. It's implied that they are in paragraph one but this is not made explicit. If the pangolins are not overexploited, i.e. they're simply being exploited, then I suppose it doesn't matter too much from a conservation perspective what exactly motivates the hunters. As stated in the paper, "African pangolins have been exploited long before being trafficked to Asia, with their exploitation tied to rural communities' use of wildlife to supplement food and income". This is so critical, in my view, for contextualizing the conservation problem here. These populations have been subject to strong hunting pressure for a long time. So, are these populations now overexploited and/or in danger of extinction? My guess is that they are... and if so, there has to be a reason underlying this. The added incentive of scales (even if secondary to meat consumption) is one potential culprit. Increased human population sizes and or demand for meat is another. We don't know. But if pangolins are threatened here then something must have changed, because presumably their palatability has not. Without knowing this, I think we need to be really careful about changing broad/global pangolin conservation strategies on the basis of these findings - otherwise we risk making the same mistakes as the authors suggest other pangolin conservationists are making at the moment.

By the way, let's look at the campaigns from conservation organizations as noted in the rebuttal (although, I generally feel that this is weak evidence for characterizing scientific consensus on issues). The WWF website states that "They certainly are one of the most trafficked mammals in Asia and, increasingly, Africa. Pangolins are in high demand in countries like China and Vietnam. Their meat is considered a delicacy and pangolin scales are used in traditional medicine and folk remedies to treat a range of ailments from asthma to rheumatism and arthritis." Based on the results here, how should this be changed to better reflect the importance of meat driving the

hunting of pangolins? I don't necessarily see this as a problematic "scales-centred narrative". The TRAFFIC website states that "As with many other wildlife species, the motivations behind consumption of pangolin products varies significantly between nations and social groups." This seems pretty consistent with the results from this study. Yes, there are some overly simplistic statements about a "global trade to feed into" as noted on the Flora & Fauna site. But I don't think this constitutes evidence for scientific consensus – there are a range of reasons for writing a statement like this on a donation page that are not science-related.

Anyway, to be clear, I'm arguing all of these points not because I don't think this work should be published or that I don't think this is important. I just hope that the findings are used in an appropriate way that effectively supports the conservation of pangolins, in Nigeria and more broadly in Africa and Asia.

I have a few suggested edits along these lines:

As raised in the rebuttal, "Additionally, our research centres on the 'relative importance or contribution' of scales and meat in driving pangolin exploitation (in other words, which is more important) – not whether one product is important and the other not." If this is the case, then I would suggest changing the title to: "Pangolin hunting in Southeast Nigeria motivated more by local meat consumption than international demand for scales". The present title definitely gives me the impression that pangolin hunting in Southeast Nigeria is "not" motivated by international demand for scales.

I would also suggest changing this Introduction sentence to:

"...there is a widespread assertion among pangolin researchers and stakeholders in Central and West Africa that international demand for scales is the primary threat faced by pangolins in these regions". This may require then better connecting the issues of "threats" and hunting "motivation".

Otherwise, the other revisions in response to my original review are good and go a long way to clarifying some of my initial concerns. The Discussion in particular is much stronger now in my opinion. I think it's pretty clear that we all agree that knowing the relative drivers of pangolin exploitation are important for management and action planning. The Discussion now well covers how these results support this.

We appreciate the clarity of your suggestions and agree with you that our results do not disprove that international demand is not threat to pangolins in SE Nigeria. We have thus accepted your suggestions as follows:

1. Revised the title to "Pangolin hunting in Southeast Nigeria motivated more by local meat consumption than international demand for scales"
2. Revised the cited text in the Introduction to "Despite this, there is a widespread view among pangolin researchers and stakeholders in Central and West Africa that international demand for pangolin scales is the primary threat to pangolins in these regions ²¹." – lines 55-57

Additional comments:

Lines 63-66 – "In a mixed scenario, where hunters primarily target pangolins for meat but trade scales as a by-product (or vice versa), the most effective interventions are likely to be those that address the primary product behind their exploitation." I disagree. I suspect the best interventions in this case would be those that address both motivations underlying the exploitation.

Thanks for this. We have clarified our statement as follows:

"In a mixed scenario, where hunters primarily target pangolins for meat but trade scales as a by-product (or vice versa), **the most effective interventions are likely those that prioritise addressing the underlying motivations driving demand for the primary product behind their exploitation**" – line 66-69

Line 210 – “may explain why”

Revised as advised.

Line 211 – again, as noted above, be careful with wording here

Revised as follows:

“The higher mass of scales compared to meat in the illegal pangolin trade^{5,7} may explain why **international demand for scales is perceived as the primary driver of African pangolin exploitation**²¹. However, our study provides four separate lines of evidence **indicating that pangolin hunting may be driven more by domestic demand for their meat.**” – lines 218-220

Lines 246-248 – As highlighted in the Introduction, pangolins here have been hunted for a long time – so what’s changed, or HAS anything changed? Is their “survival” threatened here at all?

Thanks for raising this important point. Due to the lack of robust data, we cannot discuss trends in pangolin populations in SE Nigeria. However, in line 294, we note that current rates of pangolin exploitation are likely unsustainable.

Remarks on code availability

I briefly reviewed the code and data. Everything appears to be there and appropriately annotated.

Thank you.

Reviewer #2

Many thanks for addressing my comments. I am happy with the response and suggest just two very minor points:

Thank you for your positive feedback. We have incorporated your suggestions.

line 297: "Looking beyond pangolins, our results underscore the importance of incorporating CONSIDERATION OF domestic drivers of species exploitation into DECISIONMAKING WITHIN international wildlife trade treaties". I suggest adding the capitalised text because by their very nature, international wildlife trade treaties can't generally address domestic drivers of exploitation - that generally falls within national jurisdiction. However, decision-making within these regimes can explicitly consider domestic drivers of exploitation.

Revised as advised.

Line 306: "drives" should be "drivers".

Thank you – we have corrected the typo.

Response to Reviewers (NATECOLEVOL-24092457-R2)

Reviewer #1

The authors have satisfactorily addressed my concerns, thanks. I'm sure this will be a widely read and key paper for understanding pangolin exploitation.

Thank you for your positive response and your insights, which have helped to strengthen this work.